

# The importance of mixed-phase clouds for climate sensitivity in the global aerosol-climate model ECHAM6-HAM2

Ulrike Lohmann[1] and David Neubauer[1]

[1]Institute of Atmospheric and Climate Science, ETH Zurich, Switzerland

**Correspondence:** U. Lohmann (ulrike.lohmann@env.ethz.ch)

**Abstract.** Clouds are important in the climate system because of their large influence on the radiation budget. On the one hand, they scatter solar radiation and with that cool the climate. On the other hand, they absorb and re-emit terrestrial radiation, which causes a warming. How clouds change in a warmer climate is one of the largest uncertainties for the equilibrium climate sensitivity (ECS). While a large spread in the cloud feedback arises from low-level clouds, it was recently shown that also

5    mixed-phase clouds are important for ECS. If mixed-phase clouds in the current climate contain too few supercooled cloud droplets, too much ice will change to liquid water in a warmer climate. As shown by Tan et al. (2016), this overestimates the negative cloud phase feedback and underestimates ECS in the CAM global climate model (GCM). Here we are using the newest version of the ECHAM6-HAM2 GCM to investigate the importance of mixed-phase clouds for ECS.

Although we also considerably underestimate the fraction of supercooled liquid water globally in the reference version of

10   ECHAM6-HAM2 GCM, we do not obtain increases in ECS in simulations with more supercooled liquid water in the present-day climate, contrary to the findings by Tan et al. (2016). We hypothesize that it is not the global supercooled liquid water fraction that matters, but only how well low- and mid-level mixed-phase clouds with cloud top temperatures in the mixed-phase temperature range between 0 and -35 °C are simulated. These occur most frequent in mid-latitudes, in particular over the Southern Ocean where they determine the amount of absorbed shortwave radiation. In ECHAM6-HAM2 the amount of

15   absorbed shortwave radiation over the Southern Ocean is only overestimated if all clouds below 0 °C consist exclusively of ice and only in this simulation is ECS is significantly smaller than in all other simulations. Hence, the negative cloud phase feedback seems to be important only if the optically thin low- and mid-level mid-latitude clouds have the wrong phase (ice instead of liquid water) in the absence of overlying clouds. In all other simulations, changes in cloud feedbacks associated with cloud amount and cloud top pressure, dominate.





## 1 Introduction

Changes in clouds remain one of the largest uncertainty for the calculation of the response of the climate system to a given radiative forcing $\Delta F$ (Flato et al., 2013). This response can be described with the following equation:

$$\Delta F = \Delta R + \Delta H + \lambda \Delta T_s. \tag{1}$$

Here $\Delta R$ represents the net radiative imbalance at the top-of-the-atmosphere (TOA), $\Delta H$ is the heat taken up by the ocean, $\Delta T_s$ is the net change in global annual mean surface temperature and $\lambda$ is the climate feedback parameter in $\mathrm{W\,m^{-2}\,K^{-1}}$. $\Delta R$ and $\Delta T_s$ change over time. In addition, some studies suggest that also the heat uptake by the ocean is time-dependent and even $\lambda$ is not a constant parameter (e.g., Knutti and Rugenstein, 2015).

Here we evaluate the increase in the global annual mean surface temperature ($\Delta T_s$) at the time of a doubling of carbon
dioxide ($CO_2$) with respect to pre-industrial concentrations, which amounts to $\Delta F_{2xCO2} = 3.7\,\mathrm{W\,m^{-2}}$ (Solomon et al., 2007). $\Delta T_s$ can be calculated in different ways. It can be taken from atmosphere GCM simulations coupled to a dynamic ocean at the time of $CO_2$ doubling (transient climate response, TCR, (e.g., Flato et al., 2013)) or it can be obtained from coupled atmosphere - mixed layer ocean (MLO) simulations that are abruptly exposed to a $CO_2$ doubling relative to pre-industrial concentrations and then run until a new equilibrium has been established (equilibrium climate sensitivity, ECS). While TCR is more relevant
to climate change because it is obtained from transient simulations that at the time of $CO_2$ doubling have not reached a new equilibrium climate, it is more expensive to calculate because it requires fully coupled atmosphere-ocean models. Therefore we focus on ECS in this paper. At the time of the new equilibrium in the coupled atmosphere-MLO simulations, $\Delta R$ and $\Delta H$ vanish and eq. (1) reduces to:

$$\Delta F_{2xCO2} = \lambda \Delta T_s. \tag{2}$$

and $\Delta T_s$ equals the ECS. $\lambda$ can be decomposed into different feedbacks. It is the sum of the Planck feedback, the water vapor feedback, the lapse rate feedback, the surface albedo feedback, the cloud feedback $\lambda_c$, that will be discussed further below, and a residual term. The convention is that the individual feedbacks are multiplied by -1 so the postive cloud feedback has a positive value. The residual term considers the interactions between the different components and the non-linear dependencies of ECS from the forcing (Vial et al., 2013). This term needs to be small for the decomposition into individual feedbacks
to explain the vast majority of the feedback. The average $\lambda$ value from the GCMs that participated in the Coupled Model Intercomparison Project Phase 5 (CMIP5) is $1.1\,\mathrm{W\,m^{-2}\,K^{-1}}$ with a 90% uncertainty range of $\pm 0.5\,\mathrm{W\,m^{-2}\,K^{-1}}$ (Flato et al., 2013). The uncertainty in $\lambda$ induces the uncertainty in ECS. Neither the average ECS of approximately 3 K (Collins et al., 2013) not its uncertainty has changed much over time.

In a first model intercomparison paper by Cess et al. (1989), the spread in ECS between different GCMs varied between
1.4 and 4.1 K. Estimates from the GCMs used in CMIP5 are somehow higher with 2.1 and 4.7 K (Forster et al., 2013; Flato et al., 2013), but the range in estimates remains similar. One of the prime contributor to this range in uncertainty are inter-model differences in the changes of low-level clouds (Boucher et al., 2013). While the overall cloud feedback is positiv with $0.3\,\mathrm{W\,m^{-2}\,K^{-1}}$, the spread in $\lambda_c$ is larger than that of the overall climate feedback parameter and varies by $\pm 0.7\,\mathrm{W\,m^{-2}\,K^{-1}}$





between the CMIP5 models (Flato et al., 2013). This means a negative $\lambda_c$ cannot be ruled out and the careful conclusion from the Fifth Assessment Report of the Intergovernmental Panel on Climate Change (IPCC AR5) was that $\lambda_c$ is likely (with a probability of 66%) positive. Contributors to the positive net cloud feedback are a poleward shift of the storms and decreases in the coverage of low-level clouds, both of which result in less scattering of solar radiation, as well as increases in the top of

high clouds which increase the longwave cloud radiative effect (Boucher et al., 2013).

One of the more uncertain cloud feedbacks is the negative cloud optical depth feedback. The cloud optical depth $\tau$ is proportional to the cloud water path over the effective cloud particle size. Therefore changes in $\tau$ can occur due to changes in the cloud water path, which in turn originate from changes in the hydrological cycle, changes in the size of the cloud droplets/ice crystals and changes in cloud phase from ice to water or vice versa. The negative cloud optical depth feedback is

related to a shift from cloud ice in the present-day climate to cloud liquid water at the same altitude in the warmer climate. Because cloud droplets are generally smaller and more numerous than ice crystals and they have a different refractive index, the optical depth of liquid clouds is larger than that of ice clouds for a constant cloud water path. Precipitation formation is also less efficient for liquid clouds than for ice clouds, further increasing cloud lifetime and $\tau$ of liquid clouds. If, in addition, the cloud water path increases in a warmer climate, $\tau$ will be further enlarged. All of these aspects result in a negative cloud

optical depth feedback.

Tan et al. (2016) analyzed the ratio of supercooled liquid water to the sum of cloud liquid water and cloud ice (supercooled liquid fraction, SLF) between 0 and -35 °C in the CAM5 GCM. They found SLF to be systematically underestimated with respect to CALIOP observations, i.e. too much condensate to be in the form of ice at these temperatures. This led to a too negative cloud phase feedback and in the CAM5 GCM to a too low ECS. When constraining the present-day SLF by satellite

observations, their ECS increased by up to 1.3 °C. However, as pointed out by Gettelman and Sherwood (2016), their results should be viewed by caution because the version of CAM5 used by Tan et al. (2016) did not have the most realistic climate and could have been overly sensitive to changes in mixed-phase clouds. Motivated by their study and to understand how universal the findings of Tan et al. (2016) are, here we use the ECHAM6-HAM2 GCM to calculate the impact of mixed-phase clouds on ECS.

There have been suggestions that there is an inverse relationship between ECS and the magnitude of the anthropogenic aerosol forcing (Kiehl, 2007). Such a relationship was also found in those newer CMIP5 models that match the observed temperature record (Forster et al., 2013) and underlines that uncertainties in the aerosol forcing remain a key driver of uncertainty for both TCR and ECS (e.g., Mauritsen and Pincus, 2017; Knutti et al., 2017). Lohmann (2002) found a large impact of SLF on the anthropogenic aerosol radiative forcing using the ECHAM4 GCM. Here we define the aerosol radiative forcing as the

effective radiative forcing due to aerosol-radiation and aerosol-cloud interactions (ERFari+aci, Boucher et al. (2013)). In simulations with only ice below 0 °C (ALL_ICE), ERFari+aci was only half as large as when no ice formed at temperatures above -35 °C (ALL_LIQ). This was attributed to the three times higher increase in liquid water path (LWP), which is the vertically integrated liquid water content, between pre-industrial and present-day in simulation ALL_LIQ as compared to simulation ALL_ICE. Since that study, parameterizations in the host model ECHAM and in the cloud microphysics scheme have been





improved and the one-moment aerosol scheme has been replaced by the two-moment aerosol scheme HAM (Stier et al., 2005). Therefore, we re-evaluate the impact of SLF on ERFari+aci in this paper.

## 2 Description of the model and the sensitivity studies

In this paper we use the latest version of the aerosol-climate model ECHAM6.3-HAM2.3, which assembles the most recent versions of the atmospheric general circulation model ECHAM (namely ECHAM6, as described in Stevens et al. (2013) with the most recent changes described below) and the aerosol module HAM based on the HAM2 version as described in Zhang et al. (2012), Neubauer et al. (2014) and Schultz et al. (2017) and summarized below. As our simulations are conducted with monthly mean oxidant fields instead of with interactive chemistry as in HAMMOZ (Schultz et al., 2017), we refer to it as ECHAM6-HAM2.

### 2.1 ECHAM6

ECHAM6 is the latest generation of the atmospheric general circulation model developed by the Max Planck Institute for Meteorology (Stevens et al., 2013). As its predecessors, ECHAM6 employs a spectral transform dynamical core and a flux-form semi-Lagrangian tracer transport algorithm from Lin and Rood (1996). Vertical mixing occurs through turbulent mixing, moist convection (including shallow, deep, and mid-level convection), and momentum transport by gravity waves arising from boundary effects or atmospheric disturbances. Sub-grid scale cloudiness (stratiform clouds) is represented using the scheme of Sundqvist et al. (1989), which calculates diagnostically the grid cell cloud fraction as a function of the relative humidity in the given grid cell, once a threshold value is exceeded. Cloud liquid water and cloud ice mixing ratios are treated prognostically following Lohmann and Roeckner (1996). In ECHAM6-HAM2, additional prognostic equation for the numbers concentrations of cloud droplets and ice crystals are included (Lohmann et al., 2007). The two-moment cloud microphysics scheme and the aerosol scheme are described in section 2.2. Radiative transfer in ECHAM6 is represented using the radiation transfer broadband model PSrad (Pincus and Stevens, 2013), which considers 14 and 16 bands for the shortwave (820 to 50000 cm$^{-1}$) and longwave (10 to 3000 cm$^{-1}$) parts of the spectrum, respectively (Iacono et al., 2008).

Radiative transfer is computed based on the amount of gases, aerosols and clouds in the atmosphere and their related optical properties. Trace gas concentrations of long-lived greenhouse gases are specified in the model. Optical properties of aerosol particles and clouds are pre-calculated for each band of the RRTMG scheme using Mie theory offline and read from a look-up table based on the concentration of cloud droplets and ice crystals as computed by the two-moment scheme.

ECHAM6 includes a new land-surface model JSBACH (Reick et al., 2013). JSBACH assumes that each grid box is composed of two fractions, one representing bare soil and the other being covered with vegetation, this one being further sub-divided into tiles, one for each of the 11 plant functional types distinguished in JSBACH. Soil hydrology is represented with a single-layer bucket model. A new treatment of the surface albedo (Brovkin et al., 2013) is included, which accounts for the different sections of bare and vegetated areas.





In addition to the improved representation of solar radiative transfer by the RRTMG scheme and the improved surface albedo, smaller changes are included. The vertical discretization within the troposphere (in particular in the upper troposphere and lower stratosphere) is slightly different, the representation of convective triggering has been improved, and the tuning of various model parameters was adjusted. In contrast to ECHAM5, ECHAM6 is more commonly used in a middle-atmosphere configuration, i.e., with the two verticals grids L47 and L95 that resolve the atmosphere from the surface up to 0.01 hPa (roughly 80 km).

Changes from ECHAM6.1, as coupled to HAM2 in Neubauer et al. (2014), to ECHAM6.3 used here include small changes in convection, in the middle atmosphere, concerning the seawater albedo, improvements in snow- and sea ice coverage, an improved aerosol climatology (Kinne et al., 2013) combined with a simple plume implementation of anthropogenic aerosols (Stevens et al., 2017) and an improved submodel interface. ECHAM6.3 uses the Monte-Carlo independent column approximation radiation scheme with the option for spectral sampling in time (Pincus and Stevens, 2013). The land model JSBACH has been updated with an improved hydrology and soil model. Changes in the cloud cover scheme were made to improve cloud cover in stratocumulus regions. Also, in ECHAM6.3 the artificial cloud blinking has been removed and it is now energy conserving in the physics part.

As in previous versions of ECHAM-HAM, ECHAM drives the aerosol and chemistry modules through the generic submodel interface by providing meteorological conditions such as wind, temperature, pressure, specific humidity and conditions pertaining to the land surface (taken from JSBACH) such as Leaf Area Index. Aerosol particles and their precursors are transported in the same way as water vapor and cloud species.

## 2.2 Aerosol and cloud microphysics scheme

The aerosol module HAM predicts the evolution of an aerosol ensemble considering five components: sulfate, black carbon, particulate organic matter, sea salt, and mineral dust. The aerosol spectrum is described by the superposition of seven log-normal modes ranging from 0.005 to $> 0.5\,\mu$m. Aerosol particles within a mode are assumed to be internally mixed, in the sense that each particle can consist of multiple components. Aerosols of different modes are externally mixed, meaning that they co-exist in the atmosphere as independent particles. The seven modes of the aerosol number distribution are grouped into four geometrical size classes, including the nucleation, Aitken, accumulation and coarse modes. Particles in four of the modes contain at least one soluble compound, thus they can take up water and are referred to as soluble. The particles in the other three modes consist of compounds with no or low water-solubility and are referred to as insoluble. Through aging processes, insoluble particles can become soluble. Each mode of the aerosol size number distribution is described by the three moments, the aerosol number $N$, the number median radius $r$, and the standard deviation $\sigma$. The standard deviation is assumed to be constant and is set to 1.59 for the nucleation, Aitken, and accumulation modes and to 2 for the coarse modes. The median radius of each mode is calculated from the aerosol number and aerosol mass, which are transported as tracers. For more details, please refer to the description of HAM in Stier et al. (2005) and Zhang et al. (2012).

The two-moment cloud microphysics scheme that predicts the number concentrations and mass mixing ratios of cloud droplets and ice crystals as implemented in ECHAM5 is originally described in Lohmann et al. (2007) and Lohmann and



Hoose (2009). The microphysics scheme includes all phase changes between the water components (condensation, evaporation, freezing, melting, deposition and sublimation) and precipitation processes (autoconversion of cloud droplets, accretion of raindrops with cloud droplets and snow flakes with cloud droplets and ice crystals, aggregation of ice crystals). Moreover, evaporation of raindrops and melting and sublimation of snowflakes are considered, as well as sedimentation of cloud ice. The
cloud microphysics scheme is coupled to the aerosol scheme HAM through the processes of cloud droplet activation and ice crystal nucleation (Lohmann et al., 2007) as well as through in-cloud and below-cloud scavenging (Croft et al., 2009, 2010).

    Heterogeneous freezing in mixed-phase clouds occurs by contact and immersion freezing as discussed below. In the standard configuration, cirrus clouds are assumed to form by homogeneous freezing of supercooled solution droplets (Lohmann and Kärcher, 2002). Homogeneous and heterogeneous nucleation strongly depends on the vertical velocity that results in su-
persaturation. The vertical velocity is a superposition of the grid-scale vertical velocity and a subgrid-scale component, which is linked to the turbulent kinetic energy (Lohmann and Kärcher, 2002).

    Since the ECHAM-HAM model validation in Lohmann and Hoose (2009), the cloud microphysics scheme has undergone the following scientific improvements:

1. While in the previous ECHAM6-HAM version cloud droplet activation followed the empirical scheme by Lin and
Leaitch (1997), it now uses a parameterization of cloud droplet activation based on Köhler theory by Abdul-Razzak and Ghan (2000) as discussed in Stier (2016).

2. Previously, the increase in cloud droplet number concentration (CDNC) from cloud droplets that were detrained from convective clouds, was applied everywhere in the grid box. We now weight the detrained CDNC by the detrained mass and add it to the mass-weighted CDNC of the stratiform part. In addition the split of the detrained cloud water mass into
liquid water and ice was made consistent between the number concentrations and mass mixing ratios of cloud droplets and ice crystals.

3. We now assume the shape of ice crystals to be hexagonal plates of crystal type P1a following Pruppacher and Klett (1997). Before we used the empirical mass-size relationships from their Table 2.4, which is only valid for crystals between 0.3 and 1.5 mm. We replaced that by their empirical relationship in Table 2.2, which is valid from 10 $\mu$m to
3 mm and with that covers the whole size range. Also while we assumed that all ice crystals are hexagonal plates in the cloud microphysics scheme, cirrus crystals were treated as spheres for the calculation of the effective ice crystal radius and ice cloud optical properties in Zhang et al. (2012). This inconsistency has been eliminated and ice crystals are now hexagonal plates everywhere in the module.

4. The heterogeneous nucleation scheme in the mixed-phase cloud regime between temperatures between 273.15 K and
238.15 K considers contact nucleation by mineral dust and immersion nucleation by black carbon and mineral dust following Lohmann and Hoose (2009). However because there is no evidence that Aitken mode particles contribute to freezing (Marcolli et al., 2007), we now limit the immersion freezing of black carbon to particles in the accumulation mode or larger.



5. The temperature dependence of sticking efficiency used for accretion of ice crystals by snow has been changed to the expression used in Seifert and Beheng (2006).

6. In previous versions, the minimum CDNC (CDNCmin) was set to $40\,cm^{-3}$. The justification for this choice was twofold: First, while observations of clouds with a lower CDNC exist, (e.g, Terai et al., 2014), these smaller concentrations normally occur in clouds or pockets in clouds that are much smaller than our grid boxes. Second, so far we do not account for nitrate aerosols and our treatment of secondary organic aerosols is rather simplistic and likely underestimates the organic aerosol concentration (Zhang et al., 2012). Therefore we are likely to underestimate CDNC, which we partly buffer by using CDNCmin = $40\,cm^{-3}$. However, CDNCmin has a large impact on ERFari+aci (Hoose et al., 2009). Therefore we introduced the option to have two ECHAM6.3-HAM2.3 versions, one that keeps the CDNCmin at $40\,cm^{-3}$ and one in which we lower CDNCmin to $10\,cm^{-3}$.

In addition to the scientific improvements, we removed smaller inconsistencies, such that cloud droplets/ice crystals could both grow and evaporate/sublimate in one timestep, non-zero CDNC below 238.15 K, non-zero ice crystal number concentrations above 273.15 K, inconsistencies in the calculation of cloud cover, condensation and of the ice crystal number concentration in cirrus clouds. Moreover, the two-moment cloud microphysics scheme is now energy-conserving and has been modularized.

## 3  Model set-up and experiments

In this paper, we compare results from the two release versions of ECHAM6-HAM2 the one with CDNCmin=$40\,cm^{-3}$ (simulation REF) and the one with CDNCmin=$10\,cm^{-3}$ (simulation 10/cc). Another uncertainty is related to the choice of the biomass burning emissions. It was suggested that the GFAS biomass burning emissions needed to be scaled up by a factor of 3.4 (Kaiser et al., 2012; Veira et al., 2015). They argued that this factor partly arises because of the uncertain conversion of organic carbon to organic matter and by missing ageing because interactions of aerosol particles with gas-phase species are not included in the treatment of aerosols in the ECMWF aerosol forecast model MACC. Therefore we use the GFAS biomass burning emissions as they are in our reference simulation, but perform one simulation in which these emissions are increased by a factor of 3.4 (simulation GFAS3.4).

To address the impact of SLF on ERFari+aci and ECS in ECHAM6-HAM2, we conducted one simulation with no super-cooled liquid water below 0 °C (simulation ALL_ICE) and one simulation with no ice formation at temperatures > -35 °C (simulation ALL_LIQ) similar to what has been done in Tan et al. (2016) and Lohmann (2002). Another aspect to the possible impact of ice on ERFari+aci is the nucleation in cirrus clouds. There are discussions if cirrus clouds are mainly formed by homogeneous nucleation as we assume in our reference simulation or are formed heterogeneously (Cziczo et al., 2013; Spichtinger and Krämer, 2012; Kärcher, 2017). We investigate the impact of homogeneous vs. heterogeneous freezing in cirrus clouds by performing one simulation in which all cirrus clouds form heterogeneously (simulation HET) instead of homogeneous nucleation in simulation REF. Last but not least we investigate the impact of parameterized convection on ERFari+aci



and ECS by performing a simulation in which we completely switched off convection (simulation NOCONV) as was done in Webb et al. (2015). This is normally done only in simulations run at horizontal resolutions of less than 10 km, where the vertical motions associated with deep convection start to be resolved. While our horizontal simulation is much coarser and thus a convective parameterization is needed, there are some inconsistencies in terms of microphysics between convective and

stratiform clouds. Therefore we evaluate how the cloud fields, the climate in general, ERFari+aci and ECS are simulated if only large-scale clouds are allowed to form.

All the simulations were performed in T63 spectral resolution which corresponds to $1.875\,^\circ \times 1.875\,^\circ$ and 31 vertical layers with a top at $10\,\text{hPa}$. The simulations for the calculations of ERFari+aci were run over 20 years after a three-months spin-up. To calculate ECS, ECHAM6-HAM2 has been coupled to a mixed-layer ocean (MLO). These simulations were spun up for 25

years and then run for another 25 years, over which the results were averaged.

The sensitivity studies conducted for our studies are summarized in Table 1. For the calculations of ECS, all simulations need to be in radiative equilibrium at TOA. This requires retuning. To keep the different simulations as comparable as possible, we only adjusted two parameters inside the two-moment cloud microphysics scheme. $\gamma_r$ speeds-up the autoconversion rate of cloud droplets to grow to raindrops by collision-coalescence and accounts for the missing subgrid-scale variability in cloud

water and CDNC (Wood, 2002). $\gamma_s$ is the corresponding process in the ice phase, which enhances the aggregation of ice crystals to form snow flakes (Lohmann and Ferrachat, 2010). The values of these two parameters also included in Table 1. In addition to the changes in the tuning parameters, we needed to decrease the time-step from 7.5 to 5 minutes in simulation NOCONV because of numerical stability.

**Table 1.** Set-up of the simulations together with the two tuning parameters that differ between the simulations.

| Simulation | Description | $\gamma_r$ | $\gamma_s$ |
|---|---|---|---|
| REF | Reference simulation with ECHAM6.3-HAM2.3 with CDNCmin $= 40\,\text{cm}^{-3}$ and only homogeneous freezing in cirrus clouds | 10.6 | 900 |
| 10/cc | as REF, but with CDNCmin $= 10\,\text{cm}^{-3}$ | 2.8 | 900 |
| GFAS3.4 | as REF, but with 3.4 higher biomass burning emissions | 10.6 | 900 |
| ALL_ICE | As REF, but without any supercooled liquid water at temperatures $< 0\,^\circ\text{C}$ | 4 | 900 |
| ALL_LIQ | As REF, but with only supercooled liquid water at temperatures $> -35\,^\circ\text{C}$ | 25 | 900 |
| HET | as REF, but with only heterogeneous freezing in cirrus clouds | 14 | 300 |
| NOCONV | as REF, but without any parameterization of convection | 175 | 900 |

## 4 Comparison of ECHAM6-HAM2 with observations

An overview of the annual, global mean state of the climate in the different simulations is given in Table 2 together with observations, where available. The observations of global LWP range between 30-90 g m$^{-2}$ (Stubenrauch et al., 2013; Platnick et al., 2015, 2017; Stengel et al., 2017a; Poulsen et al., 2017). Elsaesser et al. (2017) restricted the LWP retrievals to ocean



regions (LWP$_{oc}$) and estimated LWP$_{oc}$ from a combination of the SSM/I, TMI, AMSR-E, WindSat, SSMIS, AMSR-2 and GMI satellite data (MAC-LWP) as 81.4 g m$^{-2}$. Limiting the other retrievals to ocean regions yields a global LWP$_{oc}$ of 42.9 g m$^{-2}$ from MODIS (Platnick et al., 2015, 2017), 43.9 g m$^{-2}$ from ATSR2-AATSR (Stengel et al., 2017a; Poulsen et al., 2017) and 41 g m$^{-2}$ from AVHRR-PM (Stengel et al., 2017b), illustrating the huge uncertainty of estimating LWP$_{oc}$. All simulations

except for ALL_LIQ fall within this range. In simulation ALL_LIQ, LWP$_{oc}$ amounts to 110 g m$^{-2}$ because no ice is formed at temperatures > -35 °C.

The annual zonal mean LWP and ice water path (IWP) from all simulations as well as from satellite observations for LWP from multiple satellite sensors (Elsaesser et al., 2017), the MODIS satellite (Platnick et al., 2015, 2017) and the ATSR2-AATSR satellites (Stengel et al., 2017a; Poulsen et al., 2017) and IWP from Calipso/CLOUDSAT (Li et al., 2012) are shown

in Figure 1. Because the MAC-LWP observations are only available over oceans, we limit the comparison of LWP to ocean regions. Both, retrievals from visible/near infrared sensor as well as microwave sensor have biases in retrieving LWP (Seethala and Horvath, 2010; Lebsock and Su, 2014). Lebsock and Su (2014) mentioned four biases in LWP retrievals for MODIS (visible/near infrared sensor) and AMSR-E (microwave sensor): a bias at large solar zenith angle (MODIS), missing of some pixels of low-lying clouds not detected (MODIS), LWP retrievals in cloud free scenes (AMSR-E) and the partitioning of the

microwave signal between cloud and precipitation signals (AMSR-E). The LWP bias of microwave sensor based products in clear-sky scenes has been recently corrected by Elsaesser et al. (2017). The LWP estimates of microwave retrieval products are most reliable in regions where LWP is much larger than the rain water path (RWP). Thus, Elsaesser et al. (2017) recommend to restrict the LWP satellite data to regions in which LWP/(LWP+RWP) > 0.8. This LWP$_{oc,low\_pr}$ covers areas dominated by stratocumulus, tradewind cumulus regions in the subtropics, the southern ocean, high latitude in the Northern Hemisphere

and parts of north Atlantic and north Pacific. LWP$_{oc,low\_pr}$ from MAC-LWP should thus have the smallest retrieval biases. Therefore we use it to evaluate the model LWP as shown in Figure 1.

The new multi-satellite observations of LWP$_{oc}$ by Elsaesser et al. (2017) show a maximum LWP in the tropics and, on average, their estimate is twice as high as the retrievals from MODIS and AATSR. LWP$_{oc}$ from MODIS and AATSR on the contrary increases towards the poles and only a weak secondary maximum is indicated near the equator. All observations

show high values of LWP$_{oc}$ in the storm tracks on both hemispheres, but the exact location varies between the different observations. The most noticeable difference between LWP$_{oc}$ and LWP$_{oc,low\_pr}$ is the absence of data in the intertropical convergence zone (ITCZ). In fact, the agreement of simulations REF, GFAS3.4 and HET in terms of the zonal mean structure of LWP$_{oc,low\_pr}$ with observations is very good everywhere and better than with LWP$_{oc}$. LWP$_{oc,low\_pr}$ is too low at high latitudes in simulation ALL_ICE and the maxima in the extratropics are overestimated in simulations ALL_LIQ and 10/cc.

LWP$_{oc,low\_pr}$ is everywhere underestimated in simulation NOCONV, because of the large speed-up of the autoconversion rate required to reach radiative equilibrium at TOA in this set-up.

The cloud-top CDNC for oceanic clouds ($N_{l,oc,top}$) with temperatures between 268 and 300 K has also been obtained from satellite data (Bennartz and Rausch, 2017) and for temperatures > 273.2 K from the model simulations. $N_{l,oc,top}$ is less zonally symmetric than LWP as a result of the higher emissions in the Northern Hemisphere. $N_{l,oc,top}$ reaches values of > 100 cm$^{-3}$

between 30 °N and 80 °N in the observations (Figure 1). All model simulations fail to produce such high values north of 40 °N,



**Figure 1.** Annual zonal mean LWP$_{oc}$ (a), IWP (b), LWP$_{oc,low\_pr}$ (c), cloud-top CDNC of clouds between 268 and 300 K (d), SW CRE (e) and LW CRE (f) from the simulations described in Table 1 and observations of LWP from multiple satellite sensors by Elsaesser et al. (2017) (solid line), from MODIS-AQUA collection 6.1 (Platnick et al., 2015, 2017) (dotted line) and from ATSR2-AATSR (Stengel et al., 2017a) (dashed line), satellite observations of IWP from Li et al. (2012), low-precipitating oceanic LWP from Elsaesser et al. (2017), oceanic cloud-top CDNC from Bennartz and Rausch (2017), SW and LW CRE. The solid SW and LW CRE lines are from CERES (Loeb et al., 2009), the dashed ones from ERBE and the dotted one for LW CRE is from TOVS satellite data (Susskind et al., 1997).





**Table 2.** Global annual mean of oceanic $LWP_{oc}$, $LWP_{oc,low\_pr}$, IWP, vertically integrated cloud droplet globally ($N_l$), CDNC at cloud top over oceans ($N_{l,oc,top}$) and ice crystal number concentration ($N_i$), total cloud cover (CC), precipitation rate ($P$), shortwave (SW), longwave (LW) and net cloud radiative effect (CRE) from observations and the present-day model simulations with prescribed SST as described in Table 1. The satellite observations of $LWP_{oc}$, are taken from Platnick et al. (2015, 2017); Stengel et al. (2017a); Poulsen et al. (2017); Elsaesser et al. (2017), of $LWP_{oc,low\_pr}$ from Elsaesser et al. (2017), of IWP from the CloudSat/CALIPSO satellites (Li et al., 2012), of $N_{l,oc,top}$ from Bennartz and Rausch (2017) of total cloud cover from Stubenrauch et al. (2013); Matus and L'Ecuyer (2017), precipitation rate is averaged over 1981–2010 from the GPCP version 2.3 data set (Adler et al., 2003, 2012), the CRE satellite data are averaged over the period 2001–2011 from the Clouds and the Earth's Radiant Energy System Energy Balanced and Filled Ed2.6r data set (Boucher et al., 2013) and are described in (Loeb et al., 2009). For the uncertainty range of the SW CRE also data from CloudSat and CALIPSO are considered (Matus and L'Ecuyer, 2017) and for LW CRE also data from the TOVS satellite Susskind et al. (1997).

| | OBS | REF | 10/cc | GFAS3.4 | ALL_ICE | ALL_LIQ | HET | NOCONV |
|---|---|---|---|---|---|---|---|---|
| $LWP_{oc}$, $\mathrm{g\,m^{-2}}$ | 81.4 (30-90) | 70.6 | 90.0 | 72.5 | 76.1 | 111.2 | 62.6 | 53.2 |
| $LWP_{oc,low\_pr}$, $\mathrm{g\,m^{-2}}$ | 73.5±5.5 | 76.2 | 100.4 | 78.1 | 70.4 | 140.8 | 68.2 | 42.0 |
| IWP, $\mathrm{g\,m^{-2}}$ | 25±7 | 14.8 | 14.9 | 14.8 | 16.5 | 8.6 | 26 | 11.7 |
| $N_l$, $10^{10}\,\mathrm{m^{-2}}$ | ??? | 3.1 | 2.9 | 3.3 | 2.4 | 6.4 | 2.9 | 4.5 |
| $N_{l,oc,top}$, $\mathrm{cm^{-3}}$ | 72±38 | 78 | 73 | 84 | 96 | 69 | 77 | 74 |
| $N_i$, $10^8\,\mathrm{m^{-2}}$ | – | 7.9 | 8.0 | 7.9 | 8.4 | 9.3 | 10.7 | 5.8 |
| CC, % | 68±5 | 68.1 | 68.4 | 68.3 | 67.8 | 70.6 | 66.3 | 71.2 |
| $P$, $\mathrm{mm\,d^{-1}}$ | 2.7±0.2 | 2.99 | 3.00 | 2.98 | 2.93 | 2.87 | 3.04 | 3.01 |
| SW CRE, $\mathrm{W\,m^{-2}}$ | -47.3 (-44 to -53.3) | -49.9 | -49.6 | -50.2 | -48.2 | -62.6 | -50.0 | -52.2 |
| LW CRE, $\mathrm{W\,m^{-2}}$ | 26.2 (22 to 30.5) | 24.2 | 24.3 | 24.1 | 22.4 | 35.6 | 24.6 | 24.7 |
| Net CRE, $\mathrm{W\,m^{-2}}$ | -21.1 (-17.1 to -22.8) | -25.8 | -25.4 | -26.2 | -25.8 | -27 | -25.4 | -27.5 |

probably because of an insufficient transport of aerosol particles to the Arctic (Bourgeois and Bey, 2011), missing nitrate aerosols and underestimating particulate organic aerosols (Zhang et al., 2012) all of which contribute to underestimating the concentration of cloud condensation nuclei. $N_{l,oc,top}$ is well simulated south of $30\,°$N, especially in simulations 10/cc and ALL_LIQ. $N_{l,oc,top}$ is highest in simulation ALL_ICE (Table 2) and overestimated with respect to the observations. This is

5  caused by the lower speed-up of the autoconversion rate in simulation ALL_ICE (Table 1), which reduces the sinks of cloud droplets in this simulation and causes higher cloud droplet number concentrations at all altitudes as compared to simulation REF (not shown).

The zonal distribution of $N_{l,oc,top}$ in simulation NOCONV differs from the other simulations. It peaks near the equator because the clouds here cannot form by convection and instead the stratiform cloud scheme needs to take over. As the rain

10  formation depends on CDNC only in the stratiform cloud microphysics scheme but not in the convective one, warm rain is less efficiently formed in the tropics in the stratiform scheme, causing a build-up of cloud droplets in simulation NOCONV as





portrayed in Figure 1. The CDNC peak in the tropics in simulation NOCONV deviates strongly from observations as do the lower than observed values in the extratropics.

The IWP peak in the tropics is related to liquid-origin cirrus clouds (Wernli et al., 2016; Kraemer et al., 2016; Gasparini et al., 2017) forming in the anvils from deep convection in the ITCZ as well as in the warm conveyor belt in the storm tracks.

The peak in the ITCZ is not captured in any of the model simulations suggesting that the model severely underestimates the detrained cloud ice in deep convective clouds. Cloud ice in the extratropics is underestimated in all simulations but simulation HET, where the IWP is largest due to retuning as discussed above.

ECHAM6-HAM2 in general has problems to simulate IWP values in the observed range. This deficiency can largely be attributed to an underestimation in the ice crystal size in simulation REF at least for clouds with cloud optical depth < 3

(Gasparini et al., 2017). The only simulation in which IWP in the global mean agrees with the observations is simulation HET, because the heterogeneous nucleation is limited by the number concentration of dust aerosols. The dust number concentration is smaller than the number concentration of soluble aerosols that can freeze homogeneously and freezing commences at a lower relative humidity and a higher temperature in simulation HET. Because of the limited number concentration of dust aerosols and the faster depositional growth at higher temperatures, the heterogeneously nucleated ice crystals are larger than the ones

nucleated homogeneously. Because of this, they sediment and aggregate more rapidly, causing simulation HET to initially to be out of radiative balance. Here retuning was necessary to slow down the rate of aggregation and at the same time, speed up the rate of autoconversion. This resulted in one of the lowest LWPs of all simulations and highest IWP with the highest ice crystal number concentration.

The total cloud cover in ECHAM6-HAM2 is that of large-scale clouds because convective clouds are considered to be short-

lived and to decay within one model timestep except for the detrained condensate in the anvils that is taken as a source for the stratiform cloud scheme. The global mean total cloud cover does not vary much between the different simulations and it falls within the observed range of 68±5 % (Stubenrauch et al., 2013) in all simulations. It is second-largest in simulation ALL_LIQ because of its large LWP. It is highest in simulation NOCONV because here all clouds are large-scale and contribute to the cloud cover.

The observed precipitation rate from GPCP (Adler et al., 2003) of 2.7 mm d$^{-1}$ is overestimated by 6-13 % in all simulations, a feature that ECHAM6-HAM2 shares with its host model ECHAM6 (Stevens et al., 2013). As discussed in Stevens et al. (2013), GPCP seems to underreport precipitation and higher precipitation rates are more consistent with the best observational estimates of the surface energy budget (Stephens et al., 2012).

The cloud radiative effect (CRE) is the difference between the all-sky radiation at TOA and the clear-sky radiation. The

shortwave (SW) CRE amounts to -47.3 W m$^{-2}$ with a range from -46 to -53.3 W m$^{-2}$ in the satellite data (Loeb et al., 2009; Zelinka et al., 2017). All simulations, except the one with the extreme changes in SLF predict a value of about -50 W m$^{-2}$ that lies within the observed range. SW CRE is most negative and outside the observational range in simulation ALL_LIQ because of its high LWP and total cloud cover (see also Figure 1). On the contrary, it is least negative in simulation ALL_ICE, where LWP is rather small and hence $\tau$ is smallest. Here SW CRE is severely underestimated south of 40 °S due to the lack of

supercooled liquid water. While too much absorption of shortwave radiation over the southern ocean has been a problem in the



CAM5 GCM because of insufficient supercooled liquid water (Kay et al., 2016), in our model this problem only arises in the extreme simulation ALL_ICE.

LW CRE amounts to 26.2 W m$^{-2}$ with a range from 22 to 30.5 W m$^{-2}$ in the satellite data (Loeb et al., 2009; Susskind et al., 1997; Zelinka et al., 2017). As for SW CRE, all simulations but the one with the extreme assumptions in SLF predict LW CRE to be about 24-25 W m$^{-2}$ in good agreement with the observations. Again LW CRE is smallest in simulation ALL_ICE and highest and outside the observational range in simulation ALL_LIQ (see also Figure 1). Due to the severe underestimation of cloud ice in the tropics in all simulations, LW CRE is systematically underestimated in the tropics. The exception is simulation ALL_LIQ where the underestimation in tropical cloud ice is compensated for by the thickest high level clouds (Figure 2).

The global mean observed net CRE ranges between -17.1 and -22.8 W m$^{-2}$. Here all simulations are more negative than observed, i.e. they overestimate the net negative radiative effect of clouds.

The vertical distribution of the globally and annually averaged cloud liquid water and cloud ice is shown in Figure 2. Cloud liquid water has its maximum at around 800 hPa associated with low clouds where it varies between 20 and 30 mg kg$^{-1}$ in the different simulations. It is highest in simulation 10/cc because of the reduced autoconversion rate and lowest in simulation NOCONV because of the drastically enhanced autoconversion rate (Table 1). Cloud liquid water decreases to zero at around 400 hPa except in simulation ALL_LIQ where ice formation is limited to temperatures < -35 °C. Unfortunately no observational data of the vertical distribution of cloud liquid water are available.

The vertical distribution of cloud ice has been derived from Calipso/CLOUDSAT (Li et al., 2012). It peaks at 400 hPa with 6 mg kg$^{-1}$ and becomes negligible at altitudes above 100 hPa and below 900 hPa (Figure 2). Here the differences between the sensitivity experiments are more pronounced. In simulations REF, 10/cc, GFAS3.4 and NOCONV the shape of the distribution looks similar to the observed, but the peak value is underestimated by 30-50% and in simulations REF, 10/cc and GFAS3.4 a secondary maximum is present at around 780 hPa. This peak is related to cloud ice over the Southern Ocean and Antarctica (not shown). In simulation ALL_LIQ, where the global annual mean IWP is smallest due to suppressed ice formation in mixed-phase clouds, cloud ice peaks at 250 hPa and drops to zero at 600 hPa. This simulation differs most from the observations. It also has the lowest cloud cover in the lower troposphere and the highest cloud fraction between 200 and 400 hPa (Figure 2).

In simulation ALL_ICE the comparison with observations is also less favorable as compared to simulation REF because more cloud ice than observed is simulated in the lower atmosphere while the underestimation of cloud ice at higher altitudes has not noticeable improved. The overall best agreement with observations is seen in simulation HET. This is the only simulation in which the global annual mean IWP lies within the observational uncertainty (Table 2). However, the peak in cloud ice at 400 hPa is 25% too high and too narrow. Although the freezing mechanism was only changed for cirrus clouds, the overall higher amount of cloud ice extends to lower altitudes causing an overestimation in cloud ice between 600 and 900 hPa. This is mainly a result of the reduced speed-up of the aggregation rate that affects cloud ice also in mixed-phase clouds.

SLF has been obtained from the CALIOP satellite and has been compared to an earlier version of ECHAM6-HAM2 (Komurcu et al., 2014). At that time ECHAM6-HAM2 seriously underestimated SLF. This can partly be explained by differences in the definition of SLF in the satellite data and in the model. In addition, some of the model improvements mentioned above were added after the study by Komurcu et al. (2014). As the lidar signal gets attenuated at cloud optical depth (COD) > 3, the




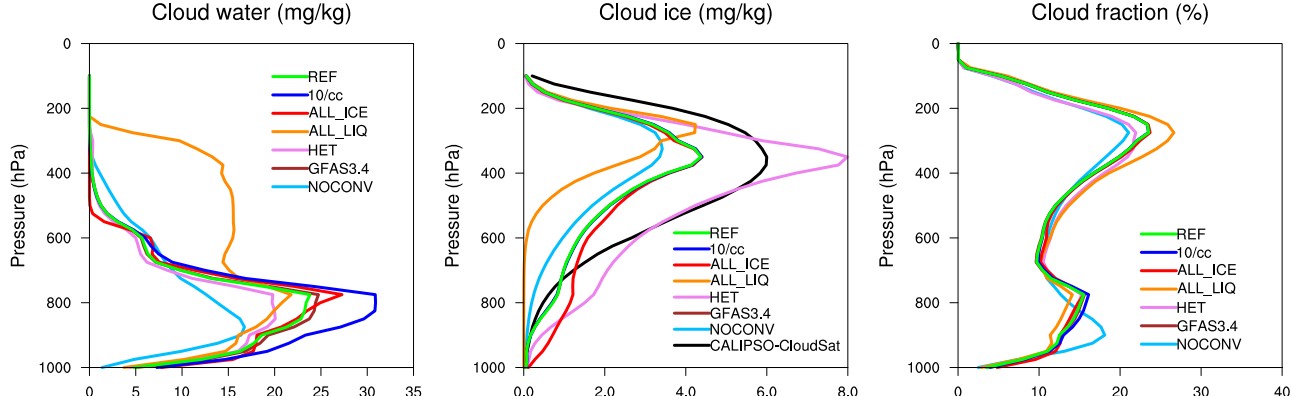

**Figure 2.** Annual global mean cloud liquid water content (left panel), ice water content (middle panel) in mg kg$^{-1}$ and cloud cover (right panel) in % as a function of pressure in hPa from the various sensitivity simulations described in Table 1 and observations of the ice water content by Li et al. (2012).

CALIOP satellite data are only representative for thin clouds with COD < 3, to which we now limit the model data as well. As shown in Figure 3, ECHAM6-HAM2 still underestimates SLF but on average simulates twice as high SLFs as presented in Komurcu et al. (2014). At -10 °C SLF amounts to 63% in CALIOP, but only 37% in simulation REF. The difference between the observed and simulated SLF decreases at colder temperatures because of the general decrease in SLF with decreasing temperature.

SLF from all ECHAM simulations is significantly underestimated except for simulations ALL_LIQ and NOCONV. Simulation NOCONV actually matches the observed SLF rather well, mainly because supercooled liquid water exists at higher altitudes than in simulation REF (Figure 2). This points to a potential deficiency as to how we handle entrainment from convective clouds or convection itself. In simulation REF we assume that if we are in the Wegener-Bergeron-Findeisen (WBF) regime, following the definition of Korolev (2007) as described in Lohmann and Hoose (2009), and cloud ice is already present, the detrained condensate will be in the form of ice. It is only detrained as supercooled liquid water if the vertical velocity is sufficiently high to exceed saturation with respect to liquid water. This assumption seems to cause a too efficient WBF process, thus depleting the supercooled liquid water too rapidly. We will rethink our approach in the future.

The control simulation in CESM from Tan et al. (2016) simulates a SLF of only 20% at -10 °C. Based on their hypothesis that an underestimation in SLF translates into an underestimation in ECS, a smaller underestimation of SLF in simulation REF should lead to a lower underestimation of ECS in ECHAM6-HAM2.

## 5 Equilibrium climate sensitivity

As mentioned above, ECS refers to the temperature that results from having established a new equilibrium climate with a balanced TOA radiation budget at the time of $CO_2$ doubling. In the warmer climate, the saturation specific humidity increases

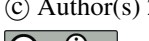



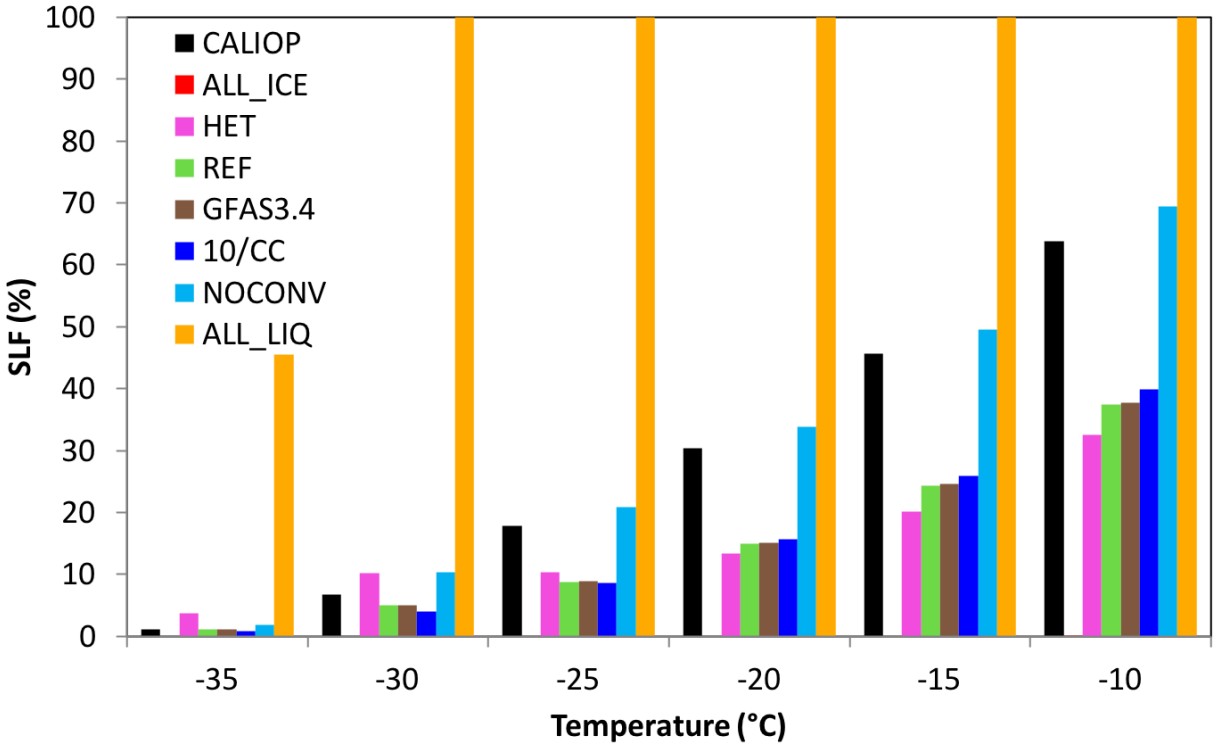

**Figure 3.** Frequency of occurrence of SLF in % for clouds with COD < 3 in different temperature bins from CALIOP observations and the various sensitivity simulations described in Table 1.

by 7% per °C warming according to the Clausius-Clapeyron equation. Because the relative humidity has been found to remain rather constant (Soden et al., 2002), this causes an increase in the specific humidity and speeds up the hydrological cycle leading to higher liquid water contents in clouds and higher precipitation rates as summarized in Table 3.

The change in ECS from all our sensitivity simulations is shown in Figure 4. ECS ranges between 1.8 and 2.8 K in all our simulations (Table 3) and with that lies on the lower side of the range of ECS estimated in the last IPCC report (Collins et al., 2013; Flato et al., 2013) and from CMIP5 models (Forster et al., 2013). Similar to Tan et al. (2016), ECS increases by ~ 30% when increasing SLF from almost zero to one (simulations ALL_ICE and ALL_LIQ in this paper vs. Low-SLF and High-SLF in Tan et al. (2016)). Contrary to their large increase of ECS from their reference simulation to simulation High-SLF, ECS does not increase between simulations REF, NOCONV and ALL_LIQ but remains at ~ 2.5 K despite the increase in SLF at -10 °C from 40% in REF to 70% in NOCONV and 100% in ALL_LIQ. Instead the ECHAM6-HAM2 simulations show a larger difference in ECS between simulations REF and ALL_ICE of 0.7 K, i.e. ECHAM6-HAM2 seems to be more sensitive at



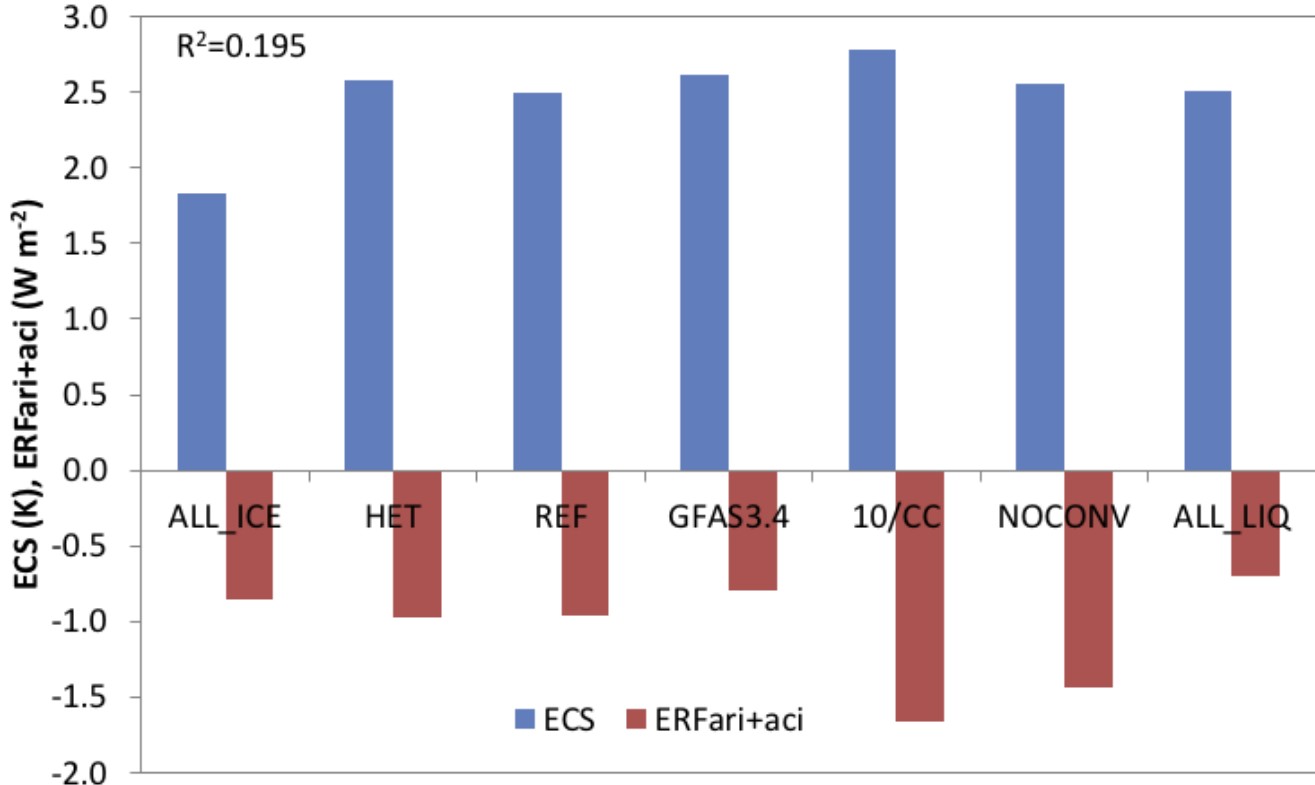

**Figure 4.** Effective radiative forcing due to aerosol-radiation and aerosol-cloud interactions (ERFari+aci) and equilibrium climate sensitivity for the various sensitivity simulations described in Table 1.

low SLF. These results seems to support the hypothesis that an increase in ECS is only sensitive to SLF if too much shortwave radiation is absorbed in mid-latitude clouds, i.e. when they are composed of ice instead of liquid water. Since in ECHAM6-HAM2 this is only the case in simulation ALL_ICE, this is the only simulation with a distinctively lower ECS. In CAM5, too much shortwave radiation is absorbed over the Southern Ocean in their present-day climate (Kay et al., 2016), which explains why ECS is also sensitive to an increase in SLF in the study by Tan et al. (2016).

In order to proof our hypothesis for the different relationship between SLF and ECS in ECHAM6-HAM2, we evaluate the changes in cloud fraction as a function of cloud top pressure and cloud optical depth between the $2xCO_2$ and $1xCO_2$ climates. Because mixed-phase clouds are more prevalent in mid- and high latitudes than in the tropics and subtropics (Matus and L'Ecuyer, 2017), the histograms in Figure 5 are taken between $20\,°\text{S}$ - $60\,°\text{S}$ and $20\,°\text{N}$ - $60\,°\text{N}$. They are taken from a satellite perspective, i.e. the cloud top pressure refers to the highest cloud in each single column. Here a phase change from cloud ice to cloud liquid water in the $2xCO_2$ climate should be most pronounced. From Figure 5 we see that only in simulation ALL_ICE there is a marked decrease of low- and mid-level optically thin extratropical clouds (with $\tau < 1.3$) accompanied by a marked increase of medium optical depth clouds ($3.6 < \tau < 60$) between 560 and 800 hPa. These changes are absent in the simulations



**Table 3.** Global annual mean changes of surface temperature ($\Delta T$), liquid water path ($\Delta$ LWP), ice water path ($\Delta$ IWP), vertically integrated cloud droplet ($\Delta N_l$) and ice crystal number concentration ($\Delta N_i$), total cloud cover ($\Delta$ CC) and precipitation rate ($\Delta P$) from the MLO model simulations described in Table 1. Different from Table 2, here the changes in LWP are global and not limited to ocean regions.

| | REF | 10/cc | GFAS3.4 | ALL_ICE | ALL_LIQ | HET | NOCONV |
|---|---|---|---|---|---|---|---|
| $\Delta T$, K | 2.5 | 2.78 | 2.61 | 1.83 | 2.51 | 2.58 | 2.56 |
| $\Delta$ LWP, $g\,m^{-2}$ | 2.54 | 2.75 | 2.59 | 4.75 | 0.80 | 3.18 | 0.72 |
| $\Delta$ IWP, $g\,m^{-2}$ | -0.40 | -0.47 | -0.45 | -0.40 | -0.54 | -1.06 | -0.31 |
| $\Delta$ CC, % | -1.43 | -1.67 | -1.53 | -0.82 | -1.13 | -1.23 | -1.15 |

with a higher SLF, here shown for REF, NOCONV and ALL_LIQ. Only in simulation ALL_ICE do these optically thin clouds consist of ice and hence are the ones that are converted to low- and mid-level liquid water clouds of medium optical depth in the warmer climate.

To estimate the radiative effect of this increase in cloud optical depth, we calculate the different components of the global
cloud feedback parameter $\lambda_c$ using the radiative kernel decomposition method described in Zelinka et al. (2013, 2016). This method decomposes $\lambda_c$ into feedbacks that are associated with changes in cloud amount $\lambda_{amt}$, cloud top pressure $\lambda_{ctp}$ and cloud optical depth $\lambda_\tau$ as shown in Figure 6 for all simulations. $\lambda_{ctp}$ is positive because of the shift of clouds to higher altitudes in the warmer climate (see also Figure 7) that enhances their LW CRE. At the same time, the total cloud amount decreases in a warmer climate, most noticeable at lower altitudes (Figure 7). This decrease in low/mid level clouds reduces the negative net
CRE and also constitutes a positive feedback. $\lambda_\tau$ is negative in most simulations and smaller than other cloud feedbacks so that $\lambda_c$ is positive in most simulations. $\lambda_c$ is negative (-0.4 W m$^{-2}$ K$^{-1}$) only in simulation ALL_ICE because of its large negative $\lambda_\tau$. In all other simulations $\lambda_c$ ranges between 0.3 W m$^{-2}$ K$^{-1}$ and 0.6 W m$^{-2}$ K$^{-1}$.

The largest differences between the simulations are associated with changes in $\lambda_\tau$, which varies between -0.6 W m$^{-2}$ K$^{-1}$ in simulation ALL_ICE and 0.1 W m$^{-2}$ K$^{-1}$ in simulation NOCONV. In simulation ALL_ICE, the large negative value of $\lambda_\tau$ can
be explained with the largest increase in the liquid water path of 4.8 g m$^{-2}$ (Table 3) due to the phase change from cloud ice to cloud liquid water mainly in low- and mid-level mid-latitude clouds (Figure 5). This causes a negative optical depth feedback in all regions, but most pronounced polewards of 40° (Figure 8). Note that simulation ALL_ICE is the only simulation in which the overall cloud feedback parameter is dominated by changes in cloud optical depth. In all other simulations, the cloud feedback is dominated by changes in cloud amount and cloud top pressure (Figure 6).
In the other simulations, shown in Figure 5 for simulations REF, NOCONV and ALL_LIQ, a rather different picture emerges. Here $\tau$ of low clouds hardly changes as the majority of them is already composed of cloud droplets in the present climate and only a small phase change from ice to liquid occurs due to the doubling of $CO_2$. In all simulations, but for simulation NOCONV, the negative cloud phase feedback shows up in the tropics, because some tropical clouds that glaciate at the tops either due to the presence of ice nucleating particles in the mixed-phase temperature regime or because their tops extend to altitudes

**Figure 5.** Changes in the distribution of cloud fraction in the extratropics (20 °S - 60 °S and 20 °N - 60 °N) as a function of cloud optical depth and cloud top pressure between the 2xCO₂ and the 1xCO₂ climate in simulations ALL_ICE (upper left), REF (upper right), NOCONV (lower left) and ALL_LIQ (lower right).




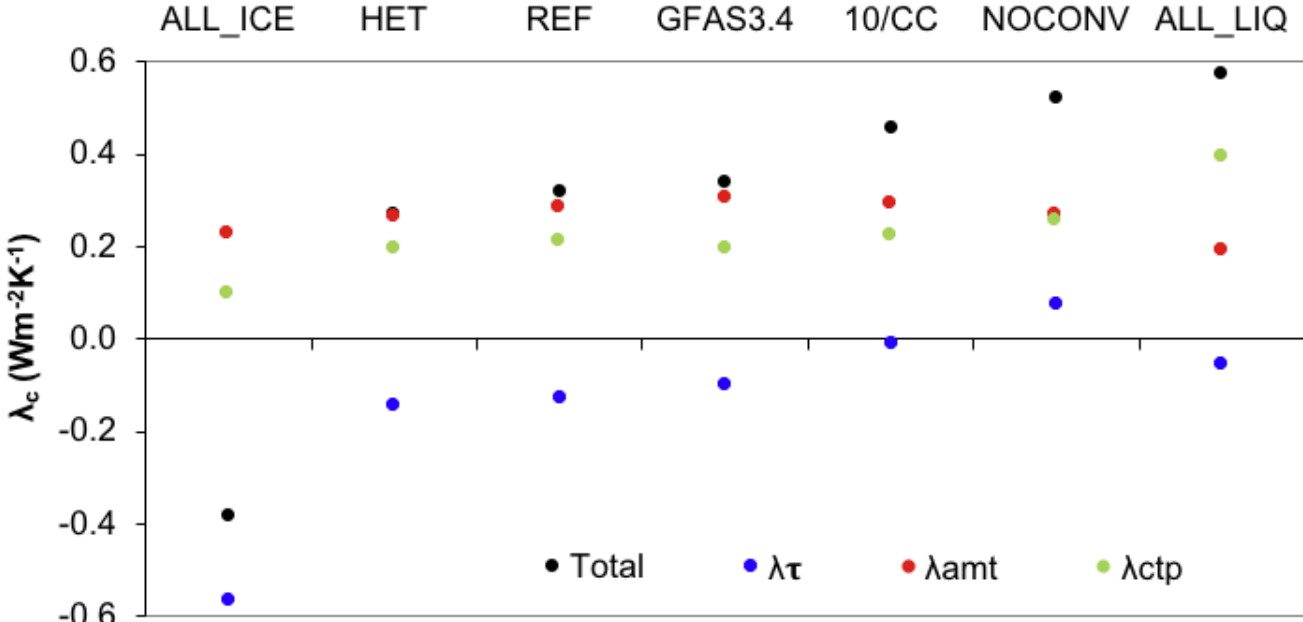

**Figure 6.** Components of the globally averaged cloud feedback parameter $\lambda_c$ in $\mathrm{W\,m^{-2}\,K^{-1}}$ for the sensitivity simulations described in Table 1. $\lambda_\tau$ accounts for changes in cloud optical depth, $\lambda_{amt}$ accounts for changes in cloud amount and $\lambda_{ctp}$ accounts for changes in cloud top pressure.

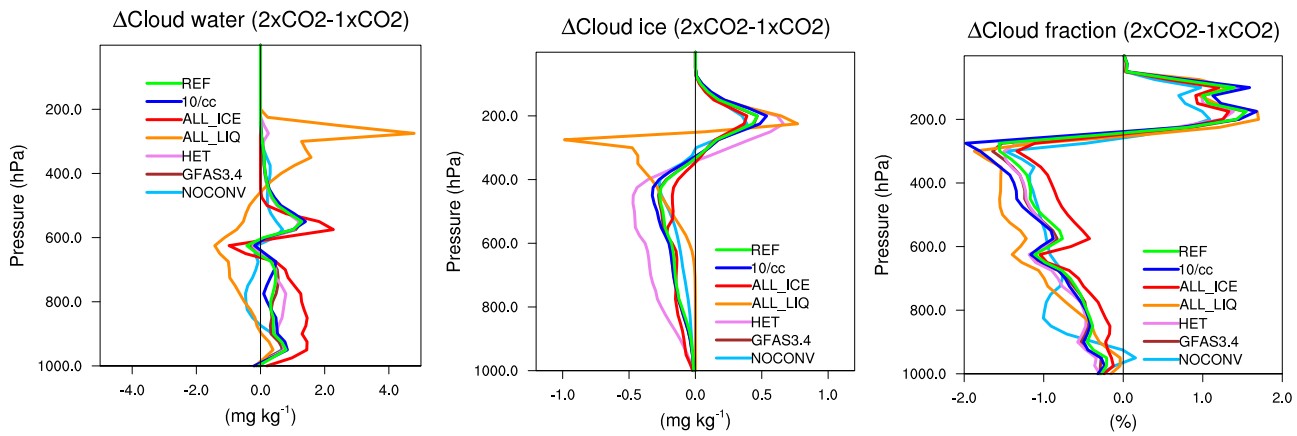

**Figure 7.** Change in the annual global mean cloud liquid water content (LWC, left panel) and ice water content (IWC, right panel) in $\mathrm{mg\,kg^{-1}}$ as a function of pressure in hPa between the $2\mathrm{xCO_2}$ and $1\mathrm{xCO_2}$ climate from the various sensitivity simulations described in Table 1.





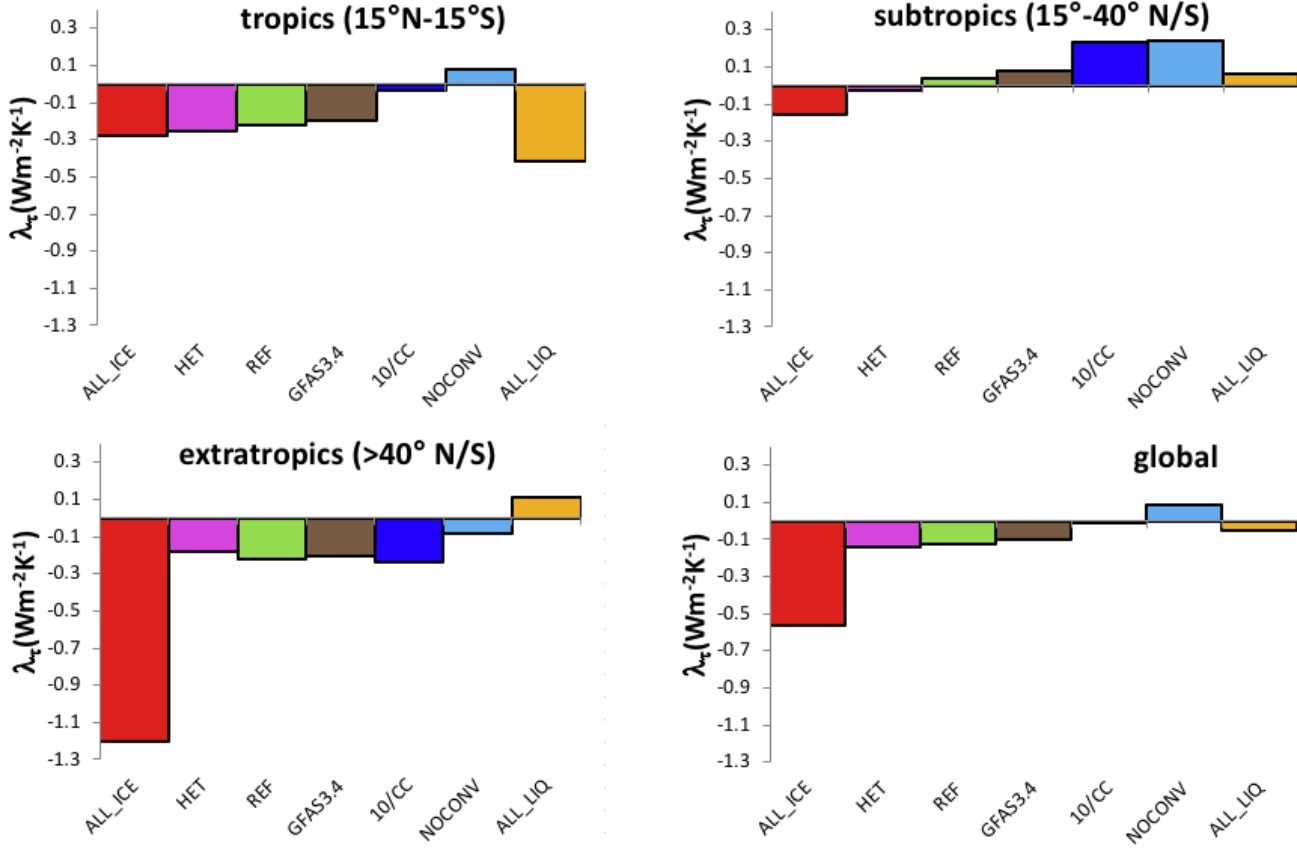

**Figure 8.** Tropical, subtropical, extratropical and global cloud optical depth feedback parameter $\lambda_\tau$ in $\mathrm{W\,m^{-2}\,K^{-1}}$ for the sensitivity simulations described in Table 1.

with temperatures $< -35\,°\mathrm{C}$ in the present climate will remain supercooled in the warmer climate and hence become optically thicker (Figure 8).

In the warmer climate, the static stability increases in areas with marine stratus, stratocumulus and tradewind cumuli in all simulations. This is accompanied by a stronger moisture gradient which promotes stronger drying by more entrainment (Gettelman and Sherwood, 2016), which in turn causes the marine subtropical clouds to thinnen and their optical depth to decrease leading in most simulations to a positive cloud optical depth feedback in this region as shown in Figure 8.

The parameterization of convective clouds in ECHAM6-HAM2 assumes that they form and dissipate in one timestep. During their lifetime, they can detrain cloud water and ice in the environment. Thus the negative cloud optical depth feedback in the tropics and mid-latitudes indicates that more cloud condensate is detrained in the form of liquid or supercooled liquid water rather than as cloud ice in the warmer climate. Thus, next to the rise of the melting level in the warmer climate also the rise of the homogeneous freezing level is important for the negative cloud phase feedback. This is best seen in simulation ALL_LIQ where no ice exists at mixed-phase temperatures, yet the negative cloud phase feedback still operates in the tropics (Figure





8). In the global mean, this negative cloud phase feedback is visible in all our simulations as a simultaneous increase in cloud liquid water and decrease in cloud ice in the warmer climate at a given pressure level (Figure 7). In mid-latitudes it shows up as an increase in optical thicker clouds and decrease in optical thinner clouds for the same cloud top pressure (Figure 5).

The second most important contributor to the differences in the overall cloud feedback between the simulations ALL_ICE
and ALL_LIQ is the change in cloud top pressure ($\lambda_{ctp}$). $\lambda_{ctp}$ varies between $0.1\,\mathrm{W\,m^{-2}\,K^{-1}}$ in simulation ALL_ICE and $0.4\,\mathrm{W\,m^{-2}\,K^{-1}}$ in simulation ALL_LIQ. The cloud top pressure feedback is mainly related to cirrus clouds and has been estimated from satellite observations to amount to $0.2\,\mathrm{W\,m^{-2}\,K^{-1}}$ (Zhou et al., 2014). We obtain a cloud top pressure feedback of about this value in all other simulations.

$\lambda_{ctp}$ is largest in simulation ALL_LIQ where the global mean changes in cloud liquid water and cloud ice are distinctively
different from all other simulations (Figure 7). It is the only simulation in which cloud liquid water decreases throughout the lower and mid troposphere because here the poleward shift of the storm tracks and the upward shift of convective clouds are least affected by changes from cloud ice to cloud liquid water. These changes in simulation ALL_LIQ are accompanied with the largest decrease in cloud cover between 750 hPa and 370 hPa. Cloud liquid water increases between 250 and 400 hPa at the expense of cloud ice in simulation ALL_LIQ. In all other simulations, the concurrent increase in cloud liquid water and
decrease in cloud ice occurs at altitudes below 400 hPa. The different vertical structure of cloud liquid water and cloud ice in simulation ALL_LIQ is a consequence of the less efficient formation of rain as compared to precipitation formation via the ice phase. Because cloud ice only forms at temperatures below -35 °C, cloud droplets are carried upwards to this temperature and the latent heat of fusion is released here. This additional buoyancy causes the highest cloud tops already in the present-day climate. In the 2xCO$_2$ climate with higher specific humidity and more latent heat release, the buoyancy is higher and causes a
further rise in cloud top pressure (see also Figure 5 for the extratropics) and the largest cloud top pressure feedback as shown in Figure 6.

The opposite effect can be seen in simulation ALL_ICE, where all cloud water is converted to ice already at 0 °C and the latent heat is released lowest in the cloud. As aggregation of ice crystals and accretion of snow flakes with ice crystals are more efficient at warmer temperatures, precipitation is formed much lower in the cloud. Therefore less condensate is carried
into higher altitudes and the increase in cloud liquid water in low and mid-level clouds in the warmer climate is largest in this simulation (Figures 5, 7 and Table 3). This combined with only a modest increase in cloud ice and cloud cover between 250 hPa and 100 hPa (Figure 7) causes the cloud top pressure feedback to be smallest in this simulation (Figure 6).

Simulation NOCONV is the only simulation in which $\lambda_\tau$ is slightly positive. $\lambda_\tau$ is negative in all other simulations in the tropics (Figure 8) because more cloud condensate is detrained in the form of liquid or supercooled liquid water in the warmer
climate. The absence of a convection parameterization and hence detrainment in simulation NOCONV on average leads to a positive tropical $\lambda_\tau$, i.e. the tropical and subtropical clouds in simulation NOCONV become optically thinner.

To summarize, the negative cloud optical depth feedback only dominates the overall cloud feedback if the low- and mid-level midlatitude clouds consist of ice in the present climate as in simulation ALL_ICE. However, this does not imply that the overall cloud feedback remains more or less constant in the other simulations. In fact, $\lambda_c$ almost doubles from simulation
REF to simulation NOCONV and ALL_LIQ, caused by the large increase in cloud top pressure in simulation ALL_LIQ and a





positive $\lambda_\tau$ in simulation NOCONV as explained above. This marked increase in the overall cloud feedback from simulations HET and REF to simulations NOCONV and ALL_LIQ, however, does not lead to an increase in ECS. We hypothesize that different reasons are responsible for this behaviour in the different simulations: In simulation ALL_LIQ, this is caused by the increase in optically thick clouds that cause convective aggregation in the ITCZ with less outgoing longwave radiation (OLR,

see Figure 9) and as seen in the most negative $\lambda_\tau$ in the tropics in this simulation (Figure 8). This convective aggregation is associated with a decrease in clouds elsewhere especially in the mid troposphere (Figure 7) from where more longwave radiation is emitted to space. This leads to an overall negative feedback (e.g., Hartmann and Larson, 2002; Mauritsen and Stevens, 2015) because of the much larger clear-sky area and offsets the positive cloud feedback limiting ECS in simulation ALL_LIQ to have the same value as in simulation REF. Whereas reductions in OLR are restricted to the tropics in most of

the simulations, in simulation ALL_ICE also less longwave radiation is emitted from clouds over the Southern Ocean due to the negative cloud phase feedback. In simulation NOCONV hardly any reduction in OLR is visible, not even in the tropics, because of the absence of convection. The associated absence of detrainment slows down the Wegener-Bergeron-Findeisen process on the one hand, but speeds up the autoconversion rate on the other hand. The latter causes the increase in cloud liquid water in the warmer climate to be smallest (Table 3), which explains the overall positive $\lambda_\tau$ and second-largest $\lambda_c$, but not why

ECS does not increase as compared to simulation REF. The absence of convection seems to limit the increase in cirrus clouds (increase in cloud cover at altitudes above 250 hPa, Figure 7), and allows more OLR to be emitted to space basically everywhere on the globe (Figure 9). Differences in increases in Arctic OLR between the sensitivity simulations point to different Arctic amplification in these simulations, but that is less important for ECS.

## 6    Aerosol radiative forcing

The aerosol radiative forcing ERFari+aci between the simulation with present-day aerosol emissions from the year 2008 and pre-industrial emissions from the year 1850 is shown in Figure 4. ERFari+aci varies between -0.7 W m$^{-2}$ in simulation ALL_LIQ and -1.7 W m$^{-2}$ in simulation 10/cc. Contrary to earlier results obtained with ECHAM4 (Lohmann, 2002), where ERFari+aci was twice as large in simulation ALL_LIQ than in simulation ALL_ICE, there is not a large difference in ER-Fari+aci between these two simulations. In Lohmann (2002), the twice as large ERFari+aci in ALL_LIQ resulted from a three

times as large LWP in the present-day and also a three times as large increase in LWP between pre-industrial and present-day conditions than in simulation ALL_ICE. In the current simulations, we retuned each simulation to be in radiative equilibrium at the top-of-the-atmosphere, which reduces the difference in LWP between these two simulations in the present-day climate to less than a factor of two.

For ERFari+aci the minimum CDNC is more important than differences in SLF. When reducing the minimum CDNC from

40 cm$^{-3}$ to 10 cm$^{-3}$, ERFari+aci increases from -1 to -1.7 W m$^{-2}$ in accordance with the findings by Lohmann et al. (2000) and Hoose et al. (2009). On the contrary, ERFari+aci decreases from -1 to -0.8 W m$^{-2}$ when the biomass burning emissions are higher as in simulation GFAS3.4 because this increases the natural background (Carslaw et al., 2013). Whether cirrus clouds are formed by homogeneous or heterogeneous nucleation (simulations HET vs. REF) has no impact on ERFari+aci.



**Figure 9.** Change in the annual mean outgoing longwave radiation in $W\,m^{-2}$ between the $2xCO_2$ and $1xCO_2$ climate from the various sensitivity simulations described in Table 1. Negative values denote higher values of OLR in the $2xCO_2$ climate.





ERFari+aci is important for the transient climate response (TCR), because a large present-day aerosol forcing implies a larger warming when greenhouse gases become more important and emissions of aerosol particles are reduced (Andreae et al., 2005). However, since TCR and ECS are related (e.g., Otto et al., 2013; Knutti et al., 2017), and TCR is affected by ERFari+aci (e.g., Ekman, 2014), also a connection between ERFari+aci and ECS seems plausible. As discussed in the introduction, such

a correlation between ERFari+aci and ECS was found in earlier versions of fully coupled atmosphere-ocean GCMs (Kiehl, 2007) and can also be seen in those newer CMIP5 models that are consistent with the observed temperature record during the instrumental period (Forster et al., 2013). However, that does not imply a causal relationship between ERFari+aci and ECS but says that GCMs which for whatever reason have a higher TCR need to have a higher ERFari+aci in order to match the observed temperature evolution.

When conducting different sensitivity simulations with a given model, a correlation between ERFari+aci and ECS can only be expected if there is a physical mechanism that affects both. In general, a doubling of $CO_2$ leads to a faster hydrological cycle which reduces the lifetime of aerosols, but it does that similarly in all simulations. In our simulations, the coefficient of determination ($r^2$) between ERFari+aci and ECS is only 0.2, i.e. no correlation exists (see Figure 4). I.e. the differences in SLF in low- and mid-level midlatitude clouds that influence ECS are not the main determinant for ERFari+aci. Vice versa, the

differences in pre-industrial climate that strongly influence ERFari+aci are secondary for ECS.

## 7   Conclusions

In this study we used the newly developed ECHAM6.3-HAM2.3 coupled global aerosol-climate model to assess the influence of different factors for ECS. This work was motivated by the findings of Tan et al. (2016) using CAM5, who showed a large influence of correcting the underestimated SLF in present-day mixed-phase clouds on ECS. An underestimate of SLF has also

been found in other models (Komurcu et al., 2014; Barrett et al., 2017a, b) and in a different version of CAM5 (Kay et al., 2016). The SLF was found to be most sensitive to the glaciation rate, which in turn is influenced by ice microphysics and the model's vertical resolution (Barrett et al., 2017a, b). In ECHAM6-HAM2, SLF could be improved when switching off the convective parameterization entirely. The absence of convection and hence detrained cloud ice slows down the glaciation rate in ECHAM6-HAM2, which corroborates the findings by Barrett et al. (2017a) on the importance of the glaciation rate for SLF.

In ECHAM6-HAM2, ECS is much smaller than in the CAM5 GCM used by Tan et al. (2016). It varies between 1.8 to 2.5 K in ECHAM6-HAM2 versus between 3.9 to 5.7 K in CAM5 (Tan et al., 2016). Thus, while ECS is at low side of the ECS range between 2.1 and 4.7 K in IPCC AR5 (Forster et al., 2013; Flato et al., 2013) in ECHAM6-HAM2, it is on the high side in CAM5. Note that the percentage increase in ECS of 30% between the extreme scenarios (ALL_ICE to ALL_LIQ in ECHAM6-HAM2 vs. Low-SLF to High-SLF in CAM5) is similar, indicating that the relative contribution of the negative

cloud phase feedback to the overall ECS is comparable in both GCMs.

However, important differences between the two GCM studies are apparent: Increasing SLF in ECHAM6-HAM2 from its variable values in the reference simulation to one as in simulation ALL_LIQ does not increase ECS in contrary to the findings by Tan et al. (2016). This supports the hypothesis put forward by Gettelman and Sherwood (2016) that the increase in ECS




by increasing SLF in the CAM5 version used by Tan et al. (2016) is overestimated because its mean climate is not the most realistic. More specifically, as discussed in Kay et al. (2016), CAM5 overestimates the shortwave absorption over the Southern Ocean because of too efficient freezing in low- and mid-level clouds. We hypothesize that it is the SLF of these optically thin low- and mid-level mid-latitude clouds in the absence of overlying clouds that matters for ECS, because the cloud top

temperature of these clouds is in the mixed-phase temperature range. Hence, the cloud phase at the cloud top of these clouds plays an important role for the TOA radiation budget. If the absorption of shortwave radiation over the Southern Ocean is correctly simulated, then SLF in other clouds does not matter for ECS. At least this is what the ECHAM6-HAM2 results show, because only in simulation ALL_ICE too much shortwave radiation is absorbed in clouds over the Southern Ocean and this is the only simulation in which ECS is significantly smaller than in all other simulations. In all other simulations sufficient or

even too little shortwave radiation is absorbed in clouds over the Southern Ocean and all of them have similar values of ECS.

The reason why an underestimation of SLF in other cloud types does not seem to matter is because the reflection of shortwave radiation saturates with increasing optical depth. In addition, there is too little shortwave radiation in higher latitudes, the tops of tropical deep convective clouds consist of ice and low-level clouds in the tropics and subtropics consist of liquid water. Thus, the cloud phase at the tops of all these other clouds will not change in the future climate. Nevertheless, it is not only

the cloud phase feedback that matters for ECS. If that were the case, ECS should be largest in simulation ALL_LIQ, where the cloud optical depth feedback parameter is highest and positive. It seems that in simulation ALL_LIQ convective clouds tend to aggregate, which causes a negative feedback by increasing the clear-sky area and with that the longwave emission from clear-sky regions to space (e.g., Hartmann and Larson, 2002; Mauritsen and Stevens, 2015). This negative feedback offsets the positive cloud feedback and causes ECS to be the same in simulations REF and ALL_LIQ. Also in simulation NOCONV,

where we switched off the convection parameterization, SLF is higher than in simulation REF, because of a reduced Wegener-Bergeron-Findeisen process. Again, ECS remains the same because of the smallest increase in cirrus cloud cover in this simulation that allows more longwave emission to space. As discussed by Frey et al. (2017), while cloud phase improvements in the extratropics affect ECS, they do not seem to matter for the warming during the 21st century in the Community Earth System Model (CESM) because of compensating responses in ocean circulation. If this is also the case in other Earth System

Models, will be subject to future investigations.

An analysis of the different simulations in terms of their aerosol radiative forcing (ERFari+aci) showed that this is largely determined by the pre-industrial aerosol concentration and hence is largest when the minimum cloud droplet number is smallest and smallest when assuming higher biomass burning emissions in pre-industrial times. No correlation has been found between SLF and ERFari+aci, contrary to earlier findings with a previous version of ECHAM-HAM (Lohmann, 2002), where the

simulations were not re-tuned to be in TOA radiation balance and hence had much larger differences in cloud water that influenced ERFari+aci. Also, no correlation has been found between ERFari+aci and ECS, suggesting that the relationship between ERFari+aci and ECS found by Forster et al. (2013) in the CMIP5 models seems to be related to structural uncertainties between different models rather than parametric uncertainties within one model.





*Acknowledgements.* The ECHAM-HAMMOZ model is developed by a consortium composed of ETH Zurich, Max Planck Institut für Meteorologie, Forschungszentrum Jülich, University of Oxford, the Finnish Meteorological Institute and the Leibniz Institute for Tropospheric Research, and managed by the Center for Climate Systems Modeling (C2SM) at ETH Zurich. The research leading to these results has received partly funding from the Center for Climate System Modelling (C2SM) at ETH Zurich, the European Union's Seventh Frame-

5    work Programme (FP7/2007-2013) project BACCHUS under grant agreement no. 603445 and from the Swiss National Science Foundation (project number 200021_160177). We like to thank Thorsten Mauritsen for useful comments and suggestions and Sylvaine Ferrachat for the help with earlier simulations on this topic.

The computing time for this work was supported by a grant from the Swiss National Supercomputing Centre (CSCS) under project ID s652 and from ETH Zurich. The data used are listed in the references and/or can be found here: CDNC climatology: Bennartz and Rausch

10   (2016), MAC-LWP: Goddard Earth Sciences Data and Information Services Center (GES DISC, current hosting: http://disc.sci.gsfc.nasa.gov, ATSR2-AASTR: Poulsen et al. (2017), AVHRR-PM: Stengel et al. (2017b), MODIS-AQUA collection 6.1: NASA Goddard (https://ladsweb.nascom.nasa.gov). Model data are available from the authors upon request.




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
