# Peer review of "The importance of mixed-phase clouds for climate sensitivity in the global aerosol-climate model ECHAM6-HAM2"

_Atmospheric Chemistry and Physics, 2018_

## Referee Comment (RC1) · Anonymous Referee #1 · 1 Mar 2018

Review of "The importance of mixed-phase clouds for climate sensitivity in the global aerosol-climate model ECHAM6-HAM2" by Lohmann and Neubauer.

Synopsis:

Lohmann and Neubauer investigate the impact of cloud phase partitioning, as quantified by supercooled liquid fraction (SLF), on equilibrium climate sensitivity (ECS) in the newest version of the ECHAM6-HAM2 GCM. Specifically, they modify the model several different ways to produce varying SLF and assess ECS for seven model configurations. They find that ECS is only strongly impacted when SLF is forced to be zero at temperatures below 0oC. This is in contrast to Tan et al. (2016) who showed

a monotonic increase in ECS with increasing SLF in the CAM5 model. The authors hypothesize that instead of global SLF, it is SLF in mid-to-high latitude clouds that is important for ECS. They find that in ECHAM6-HAM2, ECS is only impacted in the model version where absorbed shortwave radiation over the Southern Ocean is not correctly simulated. They show that ECS and ERFari+aci are not correlated in their seven realizations of ECHAH6-HAM2, in contrast to previous versions of ECHAM and the CMIP5 ensemble.

The subject matter is appropriate to ACP and the paper represents a useful addition to an ongoing area of research into the impact of cloud feedbacks on ECS. However, the current version of the paper is unclear in places, most notably the description of the model configurations and the comparison to results in Tan et al. (2016). I have two general comments and several specific comments that I recommend be addressed prior to publication.

General Comments:

1. Model Configuration. Several aspects of the model set-up and experiments section are not sufficiently explained and as a result it is unclear how the way the models were set up and run might impact the results. Specifically:

Page 7 Line 25: How are the ALL_ICE and ALL_LIQ simulations set up?

- How do the methods used to modify the model here compare to those in Tan et al. (2016)?

- Are the changes in ALL_ICE and ALL_LIQ applied to all types of clouds globally?

It was unclear to me why these seven specific configurations (other than ALL_ICE and ALL_LIQ), which contain multiple differences other than SLF, were selected given the goal of the study is to determine the impact of SLF on ECS.

Page 8 Line 7-10. It is unclear how the various simulations were set up and run. You state that "To calculate ECS, ECHAM6-HAM2 has been coupled to a mixed-layer ocean

(MLO). These simulations were spun up for 25 years and then run for another 25 years, over which the results were averaged."

- How was the mixed-layer ocean set up (i.e. how are mixed-layer depths and q-fluxes generated) for each of the seven different versions of the model?

- Are mixed-layer depth and q-fluxes different for the seven different versions of the model? If not, how might this impact results (e.g. Frey et al. (2017) showed that ocean heat uptake changes in response to differences in clouds similar to those, for example, between ALL_ICE and REF).

- How many and what types of simulations were run?

- Was a 1xCO2 (pre-industrial) simulation run for each model configuration?

- Was a 2xCO2 simulation run for each model configuration?

- When was CO2 doubled? Was it before or after the 25-year spin up period?

- You state that results were averaged over 25 years after model spin up. In the 2xCO2 case, were those 25 years all after the model had reached a new equilibrium after doubled CO2?

- Were fixed-SST runs accomplished as stated in Table 2? If so, which years? Were observed SSTs used?

- Were any fully-coupled model runs accomplished?

Table 1 could be expanded to include information to clarify many of the points above.

Page 8 Line 11-15: Model Tuning. Two parameters which impact precipitation efficiency were used to tune the various model configurations. Several places in the paper (e.g. Page 11 Lines 405, Page 12 Lines 6-7, Page 22 Lines 26-28) reference these tuning changes to explain differences between the model configurations, but a discussion of if and how tuning might impact ECS or cloud feedbacks is not included. Does model

tuning impact ECS or cloud feedbacks?

2. Comparison of results with Tan et al. (2016)

The results from this paper are contrasted with Tan et al. (2016) who showed a monotonic increase in ECS with increasing SLF in CAM5. In two places (Page 3 L20-21 and Page 25 L1-2) the authors state that the Tan et al. results should be treated with caution because the climate in their models was not "the most realistic", citing Gettelman and Sherwood (2016). This brings up two comments:

a) What about the Tan et al. (2016) model climate(s) is unrealistic that would impact ECS estimates? Are the model climates assessed in this paper more realistic? The authors compare their models to observations several ways in Table 2, but no direct comparison to Tan et al. is mentioned. The only direct comparison I could make is precipitation rate, which is similar in the Tan et al. models (their Table S2) and the model realizations presented here.

b) Gettelman and Sherwood (2016) also reference another version of CAM (Kay et al., 2016) which they state has a more realistic climate than that of Tan et al. (2016). Frey and Kay (2017) assessed the ECS of this "more realistic" version of CAM and showed an ECS increase similar to that in Tan et al. The Frey and Kay (2017) result may suggest that the Tan et al. result is not an artifact of an unrealistic climate, and should be discussed along with the Tan result.

Specific Comments:

1. Page 2 Lines 9-10: "Here we evaluate the increase in the global annual mean surface temperature ($\Delta T_s$) at the time of a doubling of carbon dioxide ($CO_2$) with respect to pre-industrial concentrations"

- This is misleading. The only warming metric assessed in this paper is ECS, which is the equilibrium global mean surface warming resulting from a doubling of $CO_2$. Not the warming at the time of doubling $CO_2$.

2. Page 2 Line 10: The forcing due to a doubling of CO2 is model-dependent (e.g. Forster et al. 2013). Is the 3.7 W m-2 listed here an average value among IPCC AR4 (Solomon et al., 2007) or the value from ECHAM6-HAM2?

3. Page 2 Lines 11-14. This discussion of TCR and ECS is misleading as it appears to imply that they are both in part defined by the model runs used to estimate them.

- "ΔTs can be calculated in different ways". This is confusing. I think you mean that there are two metrics that describe the temperature response to a doubling of CO2. TCR (warming at the time of CO2 doubling after a 1%/yr increase in a fully-coupled model) and ECS (warming at equilibrium after a doubling of CO2).

- "or it can be obtained from coupled atmosphere - mixed layer ocean (MLO) simulations that are abruptly exposed to a CO2 doubling relative to pre-industrial concentrations and then run until a new equilibrium has been established (equilibrium climate sensitivity, ECS)"

- This definition of ECS is misleading. A MLO is not necessary to estimate ECS. ECS is the global, annual mean warming at equilibrium after a doubling of CO2. It is commonly estimated using MLO models (e.g. in IPCCAR4 Meehl et al., 2007) or fully coupled models (e.g. Gregory et al., 2004; and in IPCC AR5 Flato et al., 2013).

- When comparing results to Tan et al. (2016), it is important to note that they did not use a MLO to estimate ECS. This could impact the comparison because different methods can produce differing ECS estimates (e.g. Frey et al., 2017).

4. Page 3 Lines 6-15: This paragraph should cite some of the extensive literature on the negative optical depth feedback. Some relevant papers are Mitchell et al., 1989; McCoy et al., 2015; Ceppi et al., 2016; and the review paper by Storelvmo et al, 2015.

5. Page 3 Line 25: The transition between ECS and aerosol radiative forcing could be improved. It was unclear why aerosol forcing was being discussed until the last sentence of this paragraph.

6. Page 3 Lines 30-35: Are the simulations ALL_ICE and ALL_LIQ discussed here from this paper or from a previous paper? Please clarify.

7. Page 7 Line 19: Please define the acronym GFAS.

8. Section 4 (Pages 8-14): "Comparison of ECHAM6-HAM2 with observations" This section contains a lot of information and details on the observations used. I found it hard to pick out the comparisons of the model with observations among all of these details. It might be easier to read if the information on the observations used are presented first, and then the comparison with observations is done in a more compact way.

9. Page 9 Line 35: "Nl,oc,top reaches values ofÂă > 100 cm−3 35 between 30 N and 80 N in the observations (Figure 1)." I do not see observed data north of 60 degrees North in Figure 1d.

10. Page 12 Line 6: In Figure 1 it does not appear that ALL_ICE underestimates cloud ice in the extratropics, especially in the Southern Hemisphere.

11. Page 12 Line 31 and Page 13 Line 4: Please state which simulation is "the one with the extreme changes in SLF."

12. Page 13 Lines 9-10: Does the fact that all seven model simulations overestimate the net negative radiative effect of clouds have an impact on the cloud feedbacks predicted by the models?

13. Page 14 Line 2: Here Figure 3, which shows how SLF varies between models, is introduced. Is it possible to include SLF comparisons in Table 1 and Figure 1 along with all of the other comparisons with observations?

14. Page 14 Line 19: ECS is the temperature at equilibrium after a CO2 doubling. Not "at the time of CO2 doubling"

15. Page 15 Line 3: First reference to table 3. In Table 3, how are the changes

defined? Are they the changes in response to doubled CO2? If so, which years from which simulations are used?

16. Page 16 Lines 1-5: The hypothesis put forward here is very similar to the hypothesis of Frey and Kay (2017). From Frey and Kay 2017: "Climate models overestimate the magnitude of the negative cloud phase feedback at extratropical southern latitudes because they overestimate the amount of cloud ice present in the mean state. Further, since negative feedbacks reduce warming, models with negative cloud phase feedbacks that are too large may underestimate the amount of warming resulting from greenhouse gas forcing, quantified by their equilibrium climate sensitivity."

17. Page 16 Line 9: First reference to Figure 5. In Figure 5, why is the cut off for the extratropics 60 degrees North and South? The negative cloud phase feedback acts poleward of these latitudes, especially in the Southern Hemisphere (e.g. Zelinka et al., 2012, Figure 4d).

18. Page 22 Lines 1-2: The fact that the cloud feedback increases between simulations without impacting ECS is an interesting finding. If the mixed-layer oceans are different between the different simulations, do differences in heat uptake impact this result? Are you able to assess other feedbacks (e.g. lapse rate, water vapor, surface albedo, etc.) to determine if there is compensation?

19. Page 25 Lines 12-15: This section is unclear. How is shortwave radiation at high latitudes related to the tops of deep convective clouds and low level tropical clouds? In the next sentence, what are "all of these other clouds"?

Technical Comments:

All Figures: Please specify which years from each model run (e.g. years x-y from 1xCO2 runs, years a-b from 2xCO2 runs) were used to create each figure? Which years were used for the observations presented?

Page 1 Line 13: Change "frequent" to "frequently" Page 2 Line 27: Change "not" to

"nor" Page 2 Line 30: Change "somehow" to "somewhat" Page 2 Line 31: Change "contributor" to "contributors" Page 2 Line 32: Change "positiv" to "positive" Page 8 Line 5: insert comma after ERFari+aci

Page 10 Figure 1: Label the panels a, b, c, etc. Are these model runs the MLO runs or fully-coupled runs?

Page 11 Table 2: What years are used for the observations presented here? The caption specifies some of the data sets but not all. Are the same years used in the model runs?

Page 14 Figure 2: What years are used for the observations presented here? Are the same years used in the model runs?

Page 17 Line 22: The sentence beginning on this line is very long and hard to follow. Page 20 Line 6: Is "this region" the subtropics?

References:

Ceppi P, Hartmann DL, Webb MJ (2016) Mechanisms of the negative shortwave cloud feedback in high latitudes. J Clim. doi:10.1175/JCLI-D-15-0327.1

Flato, G., Marotzke, J., Abiodun, B., Braconnot, P., Chou, S. C., Collins, W., . . . Rummukainen, M. (2013). Evaluation of climate models. In T. F. Stocker, et al. (Eds.), Climate change 2013: The physical science basis. Contribution of Working Group I to the Fifth Assessment Report of the Intergovernmental Panel on Climate Change (pp. 741–866). Cambridge, UK and New York: Cambridge University Press.

Forster, P. M., Andrews, T., Good, P., Gregory, J. M., Jackson, L. S., & Zelinka, M. (2013). Evaluating adjusted forcing and model spread for historical and future scenarios in the CMIP5 generation of climate models. Geophysical Research Letters, 118, 1139–1150. https://doi.org/10.1002/jgrd.50174

Frey, W. R., & Kay, J. E. (2017). The influence of extratropical cloud phase and amount

feedbacks on climate sensitivity. Climate Dynamics. https://doi.org/10.1007/s00382-017-3796-5

Frey, W. R., Maroon, E. A., Pendergrass, A. G., & Kay, J. E. (2017). Do Southern Ocean cloud feedbacks matter for 21st century warming? Geophysical Research Letters, 44. https://doi.org/10.1002/2017GL076339

Gregory, J. M., Ingram, W. J., Palmer, M. A., Jones, G. S., Stott, P. A., Thorpe, R. B., . . . Williams, K. D. (2004). A new method for diagnosing radiative forcing and sensitivity. Geophysical Research Letters, 31, L03205. https://doi.org/10.1029/2003GL018747

McCoy DT, Hartmann DL, Zelinka MD, Ceppi P, Grosvenor DP (2015) Mixed-phase cloud physics and Southern Ocean cloud feedback in climate models. J Geophys Res Atmos 120:9539–9554. doi:10.1002/2015JD023603

Meehl, G. A., Stocker, T. F., Collins, W. D., Friedlingstein, P., Gaye, A. T., Gregory, J. M., . . . Zhao, Z.-C. (2007). Global climate projections. In S. Solomon, et al. (Eds.), Climate change 2007: The physical science basis. Contribution of Working Group I to the Fourth Assessment Report of the Intergovernmental Panel on Climate Change (pp. 747–845). Cambridge, UK and New York: Cambridge University Press.

Mitchell JFB, Senior CA, Ingram WJ (1989) CO2 and climate: a missing feedback? Nature 341:132–134. doi:10.1038/341132a0

Storelvmo T, Tan I, Korolev AV. (2015) Cloud phase changes induced by CO2 warming—a powerful yet poorly constrained cloud-climate feedback. Curr Clim Chang Rep 1:288–296. doi:10.1007/s40641-015-0026-2

Zelinka MD, Klein SA, Hartmann DL (2012) Computing and partitioning cloud feedbacks using cloud property histograms part II: attribution to changes in cloud amount, altitude, and optical depth. J Clim 25:3736–3754. doi:10.1175/JCLI-D-11-00249.1

2018.

---

## Referee Comment (RC2) · Anonymous Referee #2 · 7 Mar 2018

Review: The importance of mixed-phase clouds for climate sensitivity in the global aerosol-climate model ECHAM6-HAM2 by Ulrike Lohmann and David Neubauer Manuscript Number acp-2018-97

In this study, the authors investigate the impact of mixed phase clouds on equilibrium climate sensitivity (ECS) and effective radiative forcing due to aerosol-radiation interactions plus aerosol-cloud interactions (ERFari+aci). Motivated by Tan et al (2016), who found that modified versions of CAM5 with higher supercooled liquid fraction (SLF) exhibited higher ECS values owing to increasingly weaker negative / stronger positive cloud optical depth feedbacks, the authors assess the robustness of this result in

similar experiments with the ECHAM6-HAM2 model. The impact of cloud phase on ECS is apparent only in comparing the reference experiment to an 'extreme' simulation with ice at all temperatures below freezing (in which case the optical depth feedback is large and negative, and the ECS is small). Experiments with progressively higher SLF show small differences in ECS, and what little change there is, is driven primarily by differences in cloud altitude feedbacks and non-cloud feedbacks. ERFari+aci values are largely insensitive to the various model formulations, but are larger in magnitude in simulations with smaller minimum cloud droplet numbers and with smaller biomass burning emissions, mimicking the effect that clouds are more susceptible to aerosol perturbations in more pristine environments.

Overall, I found the paper to be an interesting and worthwhile contribution to the literature. I have scientific concerns in a few places, and there are issues with English, so I recommend publication pending revisions.

Specific Comments

1. Page 1, Line 13: should be "most frequently"

2. Page 1, Line 19: dominate what? Inter-member spread in ECS?

3. Page 2, Line 1: "uncertainty" should be plural

4. Equation 1: $\Delta H$ should be deleted. It is not a separate term in the global TOA energy budget, and is roughly equivalent to $\Delta R$

5. Page 2, Line 9: suggest replacing "at the time of" with "in response to"

6. Introduction section: there are many references to the IPCC reports rather than to the original literature.

7. Page 2, Line 12: TCR is specifically defined for runs in which CO2 is increased 1% per year.

8. Page 2, Lines 5-20: I find it odd that there is no mention of the standard method for

computing ECS in fully coupled AOGCMs used in CMIP5: that of Gregory et al. (2004).

9. Page 2, Lines 22-23: this line about reversing the sign of feedbacks is confusing and unnecessary

10. Page 2, Line 23-24: the residual term (which does not appear in any equation, so it is unclear what it refers to), I believe, also includes errors in the kernel method. Also, the citation is missing.

11. Page 2, Line 27: "not" should be "nor"

12. Page 2, Line 31: "contributor" should be plural; "are" should be "is", or rephrase sentence appropriately

13. Page 2, Line 33: "positive" is misspelled; seems like there is a missing word(s) after "with"

14. Page 3, Lines 12-13: please provide references for this statement about precipitation efficiency

15. Page 3, Line 21: should be "viewed with caution"

16. Page 3, Line 28: strictly speaking, ECS and TCR are completely independent of aerosol forcing. Perhaps you mean observational estimates of ECS and TCR?

17. Figure 1: this has no (a), (b) labels, though the panels are referred to as such in the caption

18. Page 12, Line 15: "to initially to"

19. Page 12, Line 30: reference to Loeb should be updated to the latest EBAF product (Loeb et al. 2017)

20. Page 13, Line 27: "noticeable" should be "noticeably"

21. Page 13, Line 29: "too high" - suggest restating as "too large" so as to not confuse magnitude of the peak with vertical location in the atmosphere

22. Page 14, Line 20 - Page 15, line 3: rephrase/combine to remove redundancy

23. Page 16, Line 1: what hypothesis? Is there are reference to a paper, or to something stated earlier?

24. Page 16, Line 2: "absorbed in clouds" - do you really mean this? Or not enough SW reflected back to space by clouds?

25. Page 16, Line 6: "proof" should be "prove"

26. Page 16, Line 10: I think "single column" refers to each sub-column, as in those generated by COSP; probably need to be more clear here, and refer to COSP (Bodas-Salcedo et al. 2011)

27. Figure 5: These all look very different to me, and suggest that there are major differences in mean-state clouds as a function of CTP and tau that may be as important or more important in driving feedback differences than can be gleaned from looking at Figs 1-3 and Table 2. I suggest showing the mean-state histograms too.

28. Page 17, Lines 5: question mark before citation; also the original citations for this technique are Zelinka et al. (2012a, b), unless you are performing the decomposition separately for low clouds vs non-low clouds, in which case this citation is appropriate

29. Page 17, Line 21: suggest inserting "liquid" before "cloud droplets"

30. Page 20, Lines 3-6: it is not obvious to me that stronger entrainment drying should lead to decreased low cloud optical depth rather than low cloud coverage; also "thinnen" should be "thin".

31. Page 21, Line 3: "optical" should be "optically" (twice)

32. Page 21, Lines 4-8: is the altitude feedback computed just for free tropospheric clouds as advocated in Zelinka et al (2016), or is it done for all clouds?

33. Page 21, Line 12: should "affected" be "compensated"?

34. Page 21, Lines 9-21: It is not clear whether this paragraph is mostly speculation, or if the authors have performed analysis to convince themselves but are not showing it to the reader. The notion that the latent heat of fusion provides additional buoyancy allowing clouds to penetrate higher has been shown to be incorrect, since the atmosphere adjusts its temperature profile to closely match the temperature profile of convection (Seeley & Romps 2016). Moreover, when I look at Figure 5, I see huge CTP anomalies for thick clouds in ALL_LIQ, which are not seen for the other three simulations shown. It this because the other simulations have few clouds at these large tau values in the mean state, so there is no way of getting a strong altitude feedback, or is the upward shift truly different in the ALL_LIQ case? The change in cloud fraction profile (Figure 7) looks roughly the same in all models, so my sense is that the larger altitude feedback in ALL_LIQ comes from the fact that clouds are optically thicker (higher emissivity) in the mean-state, not something to do with how much the clouds rise in that simulation vs other simulations.

35. Page 21, line 35: "large increase in cloud top pressure" should be "large increase in cloud altitude feedback" I think.

36. Page 22, first paragraph, also page 24, lines 17-18: I found this paragraph very hard to follow, and it seems like a lot of speculation to me (though not acknowledged as such). First, models run at GCM resolution typically cannot simulate convective aggregation, so it is doubtful that that is playing a role here. Second, Figure 9 (which also lacks a caption) does not really help elucidate the processes described. A zonal mean ∆OLR figure would be a step in the right direction, so the different runs could be compared more easily. I am not aware of anything in Hartmann and Larson (2002) describing convective aggregation or a negative feedback from decreased high cloud coverage. I think the better citation is Bony et al. 2016, which relies on principles from Hartmann and Larson (2002).

37. Section 6: It is never described how ERFari+aci is computed, and with what type of experiments (fixed SSTs but modified aerosol loading, as in CMIP5?). Can the ari and

aci components be separated to get better insights about direct and indirect effects?

38. Page 24, lines 1-9: In and of itself, a large present-day aerosol forcing does not guarantee large TCR or ECS. I think you need to insert words to clarify that "given the observed change in surface temperature and ocean heat uptake, a large present-day aerosol forcing..." Similarly, TCR as strictly defined does not depend on ERFari+aci; it is simply the temperature change at the time of doubling for a simulation with CO2 increasing at 1% per year. I think you mean "TCR inferred from observations".

39. Page 24, line 11: Suggest replacing "faster" with something clearer.

40. Page 24, line 12: it is not clear what "that" refers to, or if it is shown in the paper.

41. Page 24, line 32: "variable values" is a little awkward. Also "in contrary" should be "in contrast"

42. Page 24, lines 6-7: Is this Southern Ocean bias really getting to the heart of the matter, or is it just another symptom of the fundamental issue? My understanding is that SW reflection by clouds in models whose clouds are too optically thin (e.g., due to having too low SLF) is overly sensitive to phase changes because phase changes in these models can actually have a non-negligible effect on cloud optical depth. In contrast, phase changes in in models whose mean-state clouds are not too optically thin have a negligible effect on cloud optical depth. Is this correct, and if not, could this important point be stated more clearly in the paper? It would be nice if this could be shown more unambiguously in the paper. Is it demonstrated anywhere that the models with larger SLFs actually have LWPs that put them in the range of optical depths where albedo is saturated and hence insensitive to phase changes?

43. Page 24, lines 8 and 10: "absorbed in clouds" – I don't think this is what you mean

44. Page 24, line 32: as far as I can tell, Forster et al (2013) only plots ECS against the total radiative forcing, not ERFari+aci.

45. General comment: It seems that there should be some mention of the various studies finding observational support for a model overestimate of the cloud optical depth feedback at middle latitudes (Gordon & Klein 2014, Ceppi et al. 2016, Terai et al. 2016). Currently the Tan et al study is cited alone but it is really one among several studies that are suggesting a bias, one that likely implicates the too-strong phase feedback.

References

Bodas-Salcedo A, Webb MJ, Bony S, Chepfer H, Dufresne JL, Klein SA, Zhang Y, Marchand R, Haynes JM, Pincus R, John VO (2011) COSP Satellite simulation software for model assessment. Bulletin of the American Meteorological Society 92:1023-1043

Bony S, Stevens B, Coppin D, Becker T, Reed KA, Voigt A, Medeiros B (2016) Thermodynamic control of anvil cloud amount. Proceedings of the National Academy of Sciences

Ceppi P, McCoy DT, Hartmann DL (2016) Observational evidence for a negative shortwave cloud feedback in middle to high latitudes. Geophysical Research Letters 43:1331-1339

Gordon ND, Klein SA (2014) Low-cloud optical depth feedback in climate models. Journal of Geophysical Research-Atmospheres 119:6052-6065

Gregory JM, Ingram WJ, Palmer MA, Jones GS, Stott PA, Thorpe RB, Lowe JA, Johns TC, Williams KD (2004) A new method for diagnosing radiative forcing and climate sensitivity. Geophysical Research Letters 31

Loeb NG, Doelling DR, Wang H, Su W, Nguyen C, Corbett JG, Liang L, Mitrescu C, Rose FG, Kato S (2017) Clouds and the Earth's Radiant Energy System (CERES) Energy Balanced and Filled (EBAF) Top-of-Atmosphere (TOA) Edition-4.0 Data Product. Journal of Climate 31:895-918

Seeley JT, Romps DM (2016) Tropical cloud buoyancy is the same in a world with or without ice. Geophysical Research Letters 43:3572-3579

Terai CR, Klein SA, Zelinka MD (2016) Constraining the low-cloud optical depth feedback at middle and high latitudes using satellite observations. Journal of Geophysical Research: Atmospheres:n/a-n/a

Zelinka MD, Klein SA, Hartmann DL (2012a) Computing and Partitioning Cloud Feedbacks Using Cloud Property Histograms. Part I: Cloud Radiative Kernels. Journal of Climate 25:3715-3735

Zelinka MD, Klein SA, Hartmann DL (2012b) Computing and Partitioning Cloud Feedbacks Using Cloud Property Histograms. Part II: Attribution to Changes in Cloud Amount, Altitude, and Optical Depth. Journal of Climate 25:3736-3754
* * *

---

## Referee Comment (RC3) · Anonymous Referee #3 · 18 Mar 2018

It is nice to see that more modeling groups are now looking into the issue of cloud phase and its importance for the overall cloud feedback and climate sensitivity. As such, this paper is a timely and important contribution, but it needs additional work before publication because of the following main reasons:

1) The comparison of supercooled liquid fraction (SLF) with CALIOP, which in many ways forms the foundation for the study, is not done correctly (see comment below).

2) The claim in the abstract that the findings are contrary to those of Tan et al. are not supported by the results. Tan et al. found a systematic change in equilibrium climate sensitivity (ECS) with changing cloud phase. The same relationship between cloud phase and ECS is found here, but ECHAM6 appears to lie elsewhere on the SLF scale

(has higher SLF) to begin with than the model used in Tan et al. (CESM). Many of the model experiments presented are not designed to differ from REF predominantly in their cloud phase, so they shed very little light on this relationship. In my opinion, the experiments that are relevant in this respect are ALL_ICE, REF and ALL_LIQ, but for ALL_LIQ and ALL_ICE there is at present not enough information provided on how these simulations were designed.

In addition to the above comments, I have the following additional ones (chronologically, so minor and major comments are interspersed):
Page 1, line 13: frequent should be frequently
Page 1, line 16: Remove one 'is'
Page 2, line 2: uncertainty should be uncertainties
Page 2, line 15: Emphasize that TCR is most relevant to *present-day/modern* climate change.
Page 2, lines 9-17: ECS is really the surface temperature change in response to a doubling of CO2 once the fully coupled atmosphere  ocean have reached a new equilibrium state. Therefore, simulations with an atmosphere and a MLO are only an approximation of that equilibrium state, and that should be acknowledged here. There are papers that compare the differences in equilibrium climate state depending on whether simulations are run with MLO or full ocean. It is important to point out here that the experiments in this paper differ from those of Tan et al. in this respect.
Page 2, line 29-30: It wasn't the spread that varied between 1.4 and 4.1 K, but the estimates themselves.
Page 2, line 30: Do you mean "somewhat higher" here?
Page 2, line 31: positiv should be positive
Page 3, line 3: Maybe clarify that it is the mid-latitude storms that shift poleward?
Page 3, lines 16-24: Tan et al.  used the fully coupled CESM model, not just the atmospheric CAM5.  Please clarify here what aspect of the climate of the CESM simulations in Tan et al.  was not realistic.  This was not specified in Sherwood and

Gettelman (2017), so instead of repeating their unsubstantiated claim it would be better to here explain what specifically was unrealistic in those simulations. Tan et al. also did not claim that ECS in CESM was too low (it is actually quite high), but that the cloud phase bias in isolation would bias ECS low. Other biases in CESM could potentially bias ECS high.

Page 5, line 13: What is "cloud blinking"?

Page 7, lines 25-27: How were the ALL_ICE and ALL_LIQ simulations designed? Did you switch off heterogeneous ice nucleation in ALL_LIQ? And in ALL_ICE, did you increase ice nucleation, increase ice crystal growth (through e.g. the WBF process), or both? What about detrained ice/liquid from convection, did you modify that?

Page 8, line 3: I'm guessing this should be "horizontal resolution"?

Page 11, line 35 - Page 12, line 1 2: This is a widespread problem in GCMs in general, not just in CAM5/CESM.

Page 13, line 35 -Page 14, line 1: This is not the appropriate way to sample the simulated SLF to compare with CALIOP. CALIOP measures SLF for ALL cloud tops, irrespective of their optical depth, but if the cloud top layer has an optical depth < 3 it will ALSO be able to retrieve cloud phase from the cloud layer below cloud top. That is very different from only sampling SLF from optically thin clouds (tau < 3) in the simulations, so the comparison to CALIOP in Fig. 3 is not meaningful at this point.

Page 14, line 8: Do you mean "detrainment" here? It would actually be very helpful to know exactly how ECHAM handles the cloud phase of detrained clouds.

Fig. 4 and related discussion: The simulations presented here are not comparable to the experiments in Tan et al, for the following reasons: With the exception of ALL_LIQ and ALL_ICE, the differences between REF and the other experiments go far beyond just cloud phase. Whatever difference (or lack of difference) in ECS relative to REF can therefore not be attributed to cloud phase changes alone. In addition, as pointed out above, SLF is not extracted from the simulations in an appropriate manner, so it is hard to say how SLF would compare if they had been, and since it is unclear how ALL_LIQ and ALL_ICE were constructed it is hard to make an informed comparison

with the corresponding simulations in Tan et al.

Page 16, line 6: proof should be prove.

Table 3: Give two decimals consistently in this table.

Fig. 5 shows that there are a lot of other things going on in these simulations other than the cloud phase feedback. Clearly, the ALL_LIQ cloud changes are very different from those in REF, so the fact that they have a similar ECS is the result of multiple compensating differences between them, cloud phase only being one of them.

Fig. 7 and related discussion: This figure, if anything, confirms the findings of Tan et al. if you focus on the simulations here that differ ONLY in their cloud phase. The cloud optical depth feedback changes quite dramatically (becomes less negative) from ALL_ICE to REF, and a small additional change is seen from REF to ALL_LIQ. The total change of 0.5K/Wm-2 is very strong. The main difference is that the default ECHAM6 is closer ALL_LIQ in its behavior, while CESM was more similar to the equivalent of ALL_ICE. The other experiments have lots of other changes in them beyond cloud phase and are not relevant for the discussion of the impact of cloud phase on climate sensitivity.

Page 20, line 5: thinnen should be thin

Page 21, line 3: optical should be optically

Page 22, lines 1-2: The only way that a marked increase in the overall cloud feedback can not correspond to a marked change in ECS is if either: i) The simulations had not equilibrated in the analyzed 25 year time periods, or ii) Other climate feedbacks compensate for the change in the cloud feedback. I'm guessing it's the latter, but both aspects should be addressed in the paper. In other words, what was the radiative balance for the respective experiments for the 25 yr time periods analyzed, and how did the other (non-cloud) climate feedbacks change between the experiments?

Page 24, line 31 -33: These findings are not contrary to Tan et al. - the SLF is higher in ECHAM6 than in CESM/CAM, so there is a smaller ECS increase associated with increasing SLF. It is completely consistent with Tan et al., as far as I can tell. A caveat here is obviously that in the ECHAM6 simulations SLF is not calculated they way it

has to be in order to be comparable to CALIOP, so once that's corrected the SLF comparison may look different.

---

## Short Comment (SC1) · 28 Mar 2018

Dear Prof. Lohmann,

I have enjoyed reading your interesting study, and I would just like to bring to your attention a paper that you may find relevant: http://dx.doi.org/10.1002/2018GL077217.

Best regards,

Alejandro Bodas-Salcedo
* * *
[Figure]

2018.

---

## Author Comment (AC1) · 9 Apr 2018

Thanks Alejandro for pointing us to your paper. It is useful for our study and we will be happy to refer to it.

---

## Author Comment (AC2) · 4 May 2018

Response to reviewer 1:

We thank the referee for his/her valuable comments and suggestions. We marked the responses to the comments in italics.

General Comments:

1. Model Configuration. Several aspects of the model set-up and experiments section are not sufficiently explained and as a result it is unclear how the way the models were set up and run might impact the results. Specifically:

[Figure]

Page 7 Line 25: How are the ALL_ICE and ALL_LIQ simulations set up?

The simulation ALL_ICE is set-up such that ice crystals grow at the expense of cloud droplets, which evaporate at temperatures < 0 °C, that all cloud water that is advected to colder temperatures is forced to freeze instantaneously and all the detrained cloud condensate is in the form of ice at temperatures < 0 °C. The simulation ALL_LIQ is set-up such that heterogeneous freezing is turned off in the temperature range between 0 and -35 °C and detrainment of ice crystals from convective clouds is restricted to temperatures below -35 °C. Ice crystals sedimenting from colder into warmer cloud layers melt at -35 °C. The number of cloud droplets which freeze at temperatures below -35°C had to be reduced by a factor of 100 in order to keep the ice crystal number concentration realistic in simulation ALL_LIQ. We added this description.

- How do the methods used to modify the model here compare to those in Tan et al. (2016)?

The simulations ALL_ICE and ALL_LIQ are as similar as possible as in Tan et al., 2016, but we were more thorough in strictly enforcing this set-up. However, there is one difference such that Tan et al. used results from a fully coupled simulation, while we were using a mixed-layer ocean. We added this.

- Are the changes in ALL_ICE and ALL_LIQ applied to all types of clouds globally?

The changes are applied to all layer clouds. Convective clouds in ECHAM are not radiatively active, thus we didn't change the simple microphysics of them. However, we ensured that the detrained condensate, which is a source for large-scale clouds, is detrained in the phase corresponding to the ALL_LIQ and ALL_ICE set-ups. We added this as mentioned above.

It was unclear to me why these seven specific configurations (other than ALL_ICE and ALL_LIQ), which contain multiple differences other than SLF, were selected given the goal of the study is to determine the impact of SLF on ECS.

You are right. Our primary goal is the impact of SLF on ECS. As also the second reviewer commented on this, we have deleted the section of the aerosol radiative forcing and the two simulations that were exclusively done for this, namely GFAS3.4 and 10/cc. We decided to keep the simulations HET and NOCONV because they influence the distribution of clouds and shed light into ECS.

Page 8 Line 7-10. It is unclear how the various simulations were set up and run. You state that "To calculate ECS, ECHAM6-HAM2 has been coupled to a mixed-layer ocean. (MLO). These simulations were spun up for 25 years and then run for another 25 years, over which the results were averaged."

- How was the mixed-layer ocean set up (i.e. how are mixed-layer depths and q-fluxes generated) for each of the seven different versions of the model?

A mixed-layer depth of 50 m is used in all simulations. The deep ocean heat flux Q is computed from 25 year-simulations with fixed SSTs (AMIP climatology 2000-2015), which were run for each of the seven different set-ups separately. We added that.

- Are mixed-layer depth and q-fluxes different for the seven different versions of the model? If not, how might this impact results (e.g. Frey et al. (2017) showed that ocean heat uptake changes in response to differences in clouds similar to those, for example, between ALL_ICE and REF).

Mixed-layer depths are the same (50 m) but Q-fluxes are different (see our reply to the previous question). We agree that the deep ocean heat fluxes will adjust to the different set-ups, therefore we computed them for each set-up separately.

- How many and what types of simulations were run?

All different simulations are listed in table 1. All of them were run in three different set-ups: (i) atmosphere-only simulations with prescribed SST and sea ice cover using the AMIP 2000-2015 climatology with present-day aerosol emissions and greenhouse gas concentrations, (ii) atmosphere-MLO simulation at 1xCO2 concentrations with pre-

industrial aerosol emissions and pre-industrial levels of the other greenhouse gas and (iii) as (ii) but with 2xCO2 concentrations. We added that.

- Was a 1xCO2 (pre-industrial) simulation run for each model configuration?

Yes.

- Was a 2xCO2 simulation run for each model configuration?

Yes.

- When was CO2 doubled? Was it before or after the 25-year spin up period?

The CO2 concentration was doubled before the 25-year spin up period. We added that.

- You state that results were averaged over 25 years after model spin up. In the 2xCO2 case, were those 25 years all after the model had reached a new equilibrium after doubled CO2?

Yes.

- Were fixed-SST runs accomplished as stated in Table 2? If so, which years? Were observed SSTs used?

Yes, the results in Table 2 are from 20-year atmosphere-only simulations with present-day CO2 concentrations, present-day aerosol emissions and prescribed SSTs and sea ice cover. The present-day reference year is 2008. A climatology for the years 2000-2015 of AMIP SSTs and sea ice cover was used. Greenhouse gas emissions are for the year 2008. Aerosol emissions are for the years 2003 to 2012 and after that year 2008 values were used.

- Were any fully-coupled model runs accomplished?

No.

Table 1 could be expanded to include information to clarify many of the points above.

We added the information in the main text.

Page 8 Line 11-15: Model Tuning. Two parameters which impact precipitation efficiency were used to tune the various model configurations. Several places in the paper (e.g. Page 11 Lines 405, Page 12 Lines 6-7, Page 22 Lines 26-28) reference these tuning changes to explain differences between the model configurations, but a discussion of if and how tuning might impact ECS or cloud feedbacks is not included. Does model tuning impact ECS or cloud feedbacks?

According to the study by Klocke et al. (2011), only the entrainment rate for shallow convection and the cloud mass flux above the level of neutral buoyancy noticeably influence climate sensitivity. While they did not change the conversion rate from cloud water to rain in stratiform clouds, they did that for convective clouds and found a negligible influence. Judging from our results in Figure 6, we do not expect a large influence. Testing this, would be a study on its own and is thus beyond the scope of the present study.

2. Comparison of results with Tan et al. (2016) The results from this paper are contrasted with Tan et al. (2016) who showed a monotonic increase in ECS with increasing SLF in CAM5. In two places (Page 3 L20-21 and Page 25 L1-2) the authors state that the Tan et al. results should be treated with caution because the climate in their models was not "the most realistic", citing Gettelman and Sherwood (2016). This brings up two comments:

a) What about the Tan et al. (2016) model climate(s) is unrealistic that would impact ECS estimates? Are the model climates assessed in this paper more realistic? The authors compare their models to observations several ways in Table 2, but no direct comparison to Tan et al. is mentioned. The only direct comparison I could make is precipitation rate, which is similar in the Tan et al. models (their Table S2) and the model realizations presented here.

As far as we know their clouds over the Southern Ocean consist of too much ice, i.e.

their negative cloud phase feedback is too strong. The global mean temperature in their CALIOP-SLF1 and CALIOP-SLF2 is 2-3 degrees lower than in their reference simulation. As climate feedbacks are state dependent, especially those related to ice, a colder reference climate will have implications for their estimates of ECS.

b) Gettelman and Sherwood (2016) also reference another version of CAM (Kay et al., 2016) which they state has a more realistic climate than that of Tan et al. (2016). Frey and Kay (2017) assessed the ECS of this "more realistic" version of CAM and showed an ECS increase similar to that in Tan et al. The Frey and Kay (2017) result may suggest that the Tan et al. result is not an artifact of an unrealistic climate, and should be discussed along with the Tan result.

Thanks for pointing this out. We now refer to Frey and Kay (2017) in this context and revised this paragraph in question.

Specific Comments:

1. Page 2 Lines 9-10: "Here we evaluate the increase in the global annual mean surface temperature ($\Delta Ts$) at the time of a doubling of carbon dioxide ($CO_2$) with respect to pre-industrial concentrations"

- This is misleading. The only warming metric assessed in this paper is ECS, which is the equilibrium global mean surface warming resulting from a doubling of $CO_2$. Not the warming at the time of doubling $CO_2$.

We changed that.

2. Page 2 Line 10: The forcing due to a doubling of $CO_2$ is model-dependent (e.g. Forster et al. 2013). Is the 3.7 W m-2 listed here an average value among IPCC AR4 (Solomon et al., 2007) or the value from ECHAM6-HAM2?

We used this value just to indicate the magnitude of the $CO_2$ forcing. We know mention that it is an average value from the IPCC AR4 models.

3. Page 2 Lines 11-14. This discussion of TCR and ECS is misleading as it appears to imply that they are both in part defined by the model runs used to estimate them.

- "ΔTs can be calculated in different ways". This is confusing. I think you mean that there are two metrics that describe the temperature response to a doubling of CO2. TCR (warming at the time of CO2 doubling after a 1%/yr increase in a fully-coupled model) and ECS (warming at equilibrium after a doubling of CO2).

We changed that.

- "or it can be obtained from coupled atmosphere - mixed layer ocean (MLO) simulations that are abruptly exposed to a CO2 doubling relative to pre-industrial concentrations and then run until a new equilibrium has been established (equilibrium climate sensitivity, ECS)"

- This definition of ECS is misleading. A MLO is not necessary to estimate ECS. ECS is the global, annual mean warming at equilibrium after a doubling of CO2. It is commonly estimated using MLO models (e.g. in IPCCAR4 Meehl et al., 2007) or fully coupled models (e.g. Gregory et al., 2004; and in IPCC AR5 Flato et al., 2013).

We changed that.

- When comparing results to Tan et al. (2016), it is important to note that they did not use a MLO to estimate ECS. This could impact the comparison because different methods can produce differing ECS estimates (e.g. Frey et al., 2017).

Thanks for pointing this out. We now refer to Frey et al., 2017 for this.

4. Page 3 Lines 6-15: This paragraph should cite some of the extensive literature on the negative optical depth feedback. Some relevant papers are Mitchell et al., 1989; McCoy et al., 2015; Ceppi et al., 2016; and the review paper by Storelvmo et al, 2015.

Thanks for the references, we added some.

5. Page 3 Line 25: The transition between ECS and aerosol radiative forcing could

be improved. It was unclear why aerosol forcing was being discussed until the last sentence of this paragraph.

As mentioned above, we deleted the entire section to make the paper more concise.

6. Page 3 Lines 30-35: Are the simulations ALL_ICE and ALL_LIQ discussed here from this paper or from a previous paper? Please clarify.

They are from the Lohmann (2002) study. We made that clearer.

7. Page 7 Line 19: Please define the acronym GFAS.

Not relevant any longer as we deleted this simulation.

8. Section 4 (Pages 8-14): "Comparison of ECHAM6-HAM2 with observations" This section contains a lot of information and details on the observations used. I found it hard to pick out the comparisons of the model with observations among all of these details. It might be easier to read if the information on the observations used are presented first, and then the comparison with observations is done in a more compact way.

We prefer to keep the description of the observations where they are introduced. A summary of the different observations can be found in Table 2 and Figure 1.

9. Page 9 Line 35: "Nl,oc,top reaches values of > 100 cm-3 between 30 N and 80 N in the observations (Figure 1)." I do not see observed data north of 60 degrees North in Figure 1d.

Thanks for spotting this. We corrected that.

10. Page 12 Line 6: In Figure 1 it does not appear that ALL_ICE underestimates cloud ice in the extratropics, especially in the Southern Hemisphere.

You are right. We corrected that.

11. Page 12 Line 31 and Page 13 Line 4: Please state which simulation is "the one

with the extreme changes in SLF."

We now explicitly refer to simulation ALL_LIQ.

12. Page 13 Lines 9-10: Does the fact that all seven model simulations overestimate the net negative radiative effect of clouds have an impact on the cloud feedbacks predicted by the models?

That is a good question. We do not have any simulations in which the net CRE is smaller so we have no way to answer this question. We would speculate similarly as we did for the last question of your major comments, that we do not anticipate a large sensitivity because our model is generally not that sensitive to model changes as seen in Figure 6.

13. Page 14 Line 2: Here Figure 3, which shows how SLF varies between models, is introduced. Is it possible to include SLF comparisons in Table 1 and Figure 1 along with all of the other comparisons with observations?

This would be possible but we find this visual comparison more appealing than adding 6 rows to Table 1. We prefer to keep this figure separately.

14. Page 14 Line 19: ECS is the temperature at equilibrium after a CO2 doubling. Not "at the time of CO2 doubling"

We corrected that.

15. Page 15 Line 3: First reference to table 3. In Table 3, how are the changes defined? Are they the changes in response to doubled CO2? If so, which years from which simulations are used?

Yes, these are the changes in response to CO2 doubling from the years 26-50. We added this in the table caption.

16. Page 16 Lines 1-5: The hypothesis put forward here is very similar to the hypothesis of Frey and Kay (2017). From Frey and Kay 2017: "Climate models overestimate
the magnitude of the negative cloud phase feedback at extratropical southern latitudes because they overestimate the amount of cloud ice present in the mean state. Further, since negative feedbacks reduce warming, models with negative cloud phase feedbacks that are too large may underestimate the amount of warming resulting from greenhouse gas forcing, quantified by their equilibrium climate sensitivity."

Thanks for pointing us to this paper, we now cite them when discussing our hypothesis.

17. Page 16 Line 9: First reference to Figure 5. In Figure 5, why is the cut off for the extratropics 60 degrees North and South? The negative cloud phase feedback acts poleward of these latitudes, especially in the Southern Hemisphere (e.g. Zelinka et al., 2012, Figure 4d).

For consistency, we now show Figure 5 polewards of 40°.

18. Page 22 Lines 1-2: The fact that the cloud feedback increases between simulations without impacting ECS is an interesting finding. If the mixed-layer oceans are different between the different simulations, do differences in heat uptake impact this result? Are you able to assess other feedbacks (e.g. lapse rate, water vapor, surface albedo, etc.) to determine if there is compensation?

We did not diagnose any other feedbacks and because of this need to speculate about the compensating processes. The heat uptake by the ocean is only relevant for TCR because it vanishes in equilibrium as we discuss in the introduction.

19. Page 25 Lines 12-15: This section is unclear. How is shortwave radiation at high latitudes related to the tops of deep convective clouds and low level tropical clouds? In the next sentence, what are "all of these other clouds"?

We want to make the point that only mixed-phase clouds with tops between 0 and -35°C matter for climate sensitivity but arguing why all other cloud types do not matter. We will make that clearer.

Technical Comments: All Figures: Please specify which years from each model run

(e.g. years x-y from 1xCO2 runs, years a-b from 2xCO2 runs) were used to create each figure? Which years were used for the observations presented?

We added the details in all figures.

Page 1 Line 13: Change "frequent" to "frequently" Page 2 Line 27: Change "not" to nor" Page 2 Line 30: Change "somehow" to "somewhat" Page 2 Line 31: Change "contributor" to "contributors" Page 2 Line 32: Change "positiv" to "positive" Page 8 Line 5: insert comma after ERFari+aci

These typos have been corrected.

Page 10 Figure 1: Label the panels a, b, c, etc. Are these model runs the MLO runs or fully-coupled runs?

The panels are now labeled with a,b,c,etc. The results are from the atmosphere-only simulations for the present-day climate. We will add that.

Page 11 Table 2: What years are used for the observations presented here? The caption specifies some of the data sets but not all. Are the same years used in the model runs?

We added the details about the observations in Figure 1 and refer to Figure 1 for this. The majority of the observations is averaged over 2003-2012. Therefore, we also use the average over the years 2003-2012 from the atmospheric-only present-day simulations for this comparison.

Page 14 Figure 2: What years are used for the observations presented here? Are the same years used in the model runs?

As mentioned above, we added that.

Page 17 Line 22: The sentence beginning on this line is very long and hard to follow.

We rewrote this sentence and split it in two.

Page 20 Line 6: Is "this region" the subtropics?

Yes, we now say that explicitly.

References:

Klocke, D., R. Pincus, and J. Quaas (2011), On constraining estimates of climate sensitivity with present-day observations through model weighting, J. Clim., 24(23), 6092–6099, doi:10.1175/2011JCLI4193.1.

---

## Author Comment (AC4) · 4 May 2018

We thank the referee for his/her valuable comments and suggestions. We marked the responses to the comments in red. Specific Comments

1. Page 1, Line 13: should be "most frequently"

Changed

2. Page 1, Line 19: dominate what? Inter-member spread in ECS?

Other feedbacks dominate the overall cloud feedback. We specified that.

[Figure]

3. Page 2, Line 1: "uncertainty" should be plural

Changed

4. Equation 1: $\Delta H$ should be deleted. It is not a separate term in the global TOA energy budget, and is roughly equivalent to $\Delta R$

We would argue that $\Delta H$ matters for the global TOA energy budget. Before equation (2), we acknowledge that it vanishes when a new equilibrium is reached. We prefer to keep it as is.

5. Page 2, Line 9: suggest replacing "at the time of" with "in response to"

Changed

6. Introduction section: there are many references to the IPCC reports rather than to the original literature.

We now refer to the original papers by Dufresne and Bony (2008) and to Yokohata et al. (2010).

7. Page 2, Line 12: TCR is specifically defined for runs in which CO2 is increased 1

We added that.

8. Page 2, Lines 5-20: I find it odd that there is no mention of the standard method for computing ECS in fully coupled AOGCMs used in CMIP5: that of Gregory et al. (2004).

We now mention the Gregory method.

9. Page 2, Lines 22-23: this line about reversing the sign of feedbacks is confusing and unnecessary

We deleted this sentence.

10. Page 2, Line 23-24: the residual term (which does not appear in any equation, so it is unclear what it refers to), I believe, also includes errors in the kernel method. Also,

the citation is missing.

We added that the residual term also includes errors in the kernel method and added references to Soden et al. (2008) and Shell et al. (2008).

11. Page 2, Line 27: "not" should be "nor"

Corrected

12. Page 2, Line 31: "contributor" should be plural; "are" should be "is", or rephrase sentence appropriately

Corrected

13. Page 2, Line 33: "positive" is misspelled; seems like there is a missing word(s) after "with"

Corrected

14. Page 3, Lines 12-13: please provide references for this statement about precipitation efficiency

We added references.

15. Page 3, Line 21: should be "viewed with caution"

Corrected

16. Page 3, Line 28: strictly speaking, ECS and TCR are completely independent of aerosol forcing. Perhaps you mean observational estimates of ECS and TCR?

We deleted everything related to aerosol forcing to make the paper more precise.

17. Figure 1: this has no (a), (b) labels, though the panels are referred to as such in the caption

Labels have been be added to Figure 1.

18. Page 12, Line 15: "to initially to"

Corrected

19. Page 12, Line 30: reference to Loeb should be updated to the latest EBAF product (Loeb et al. 2017)

We added reference to Loeb et al. (2018).

20. Page 13, Line 27: "noticeable" should be "noticeably"

Corrected

21. Page 13, Line 29: "too high" - suggest restating as "too large" so as to not confuse magnitude of the peak with vertical location in the atmosphere

Corrected

22. Page 14, Line 20 - Page 15, line 3: rephrase/combine to remove redundancy

The redundant sentence has been removed.

23. Page 16, Line 1: what hypothesis? Is there are reference to a paper, or to something stated earlier?

The hypothesis is explained in the abstract. We added a more thorough description here and referenced two new studies on this topic (Frey and Kay, 2017 and Bodas-Salcedo, 2018).

24. Page 16, Line 2: "absorbed in clouds" - do you really mean this? Or not enough SW reflected back to space by clouds?

You are right, we meant not sufficient SW is reflected back to space, we changed that.

25. Page 16, Line 6: "proof" should be "prove"

Corrected

26. Page 16, Line 10: I think "single column" refers to each sub-column, as in those generated by COSP; probably need to be more clear here, and refer to COSP (Bodas-Salcedo et al. 2011)

We added the reference to COSP-ISCCP and Bodas-Salcedo et al. (2011) and refer to the sub-columns of the simulator.

27. Figure 5: These all look very different to me, and suggest that there are major differences in mean-state clouds as a function of CTP and tau that may be as important or more important in driving feedback differences than can be gleaned from looking at Figs 1-3 and Table 2. I suggest showing the mean-state histograms too.

We added mean-state histrograms.

28. Page 17, Lines 5: question mark before citation; also the original citations for this technique are Zelinka et al. (2012a, b), unless you are performing the decomposition separately for low clouds vs non-low clouds, in which case this citation is appropriate

We do not see a question mark....We changed the citation to Zelinka et al. 2012a,b because we don't look at the differences in low vs. non-low clouds.

29. Page 17, Line 21: suggest inserting "liquid" before "cloud droplets"

Liquid cloud droplets would be redundant because cloud droplets are always liquid. Therefore, we kept cloud droplets.

30. Page 20, Lines 3-6: it is not obvious to me that stronger entrainment drying should lead to decreased low cloud optical depth rather than low cloud coverage; also "thinnen" should be "thin".

A decrease in cloud coverage will also lead to a decrease in cloud optical depth. We kept that but corrected "thinnen" to "become thinner".

31. Page 21, Line 3: "optical" should be "optically" (twice)

[Figure]

Corrected

32. Page 21, Lines 4-8: is the altitude feedback computed just for free tropospheric clouds as advocated in Zelinka et al (2016), or is it done for all clouds?

We used both methods (feedbacks calculated for all clouds and for low and non-low clouds separately) and found qualitatively similar results. For simplicity we use the feedbacks from all clouds but mention new that the decomposition by low and non-low clouds gives qualitatively similar results.

33. Page 21, Line 12: should "affected" be "compensated"?

We think that both adjectives would do, but compensated sounds more precise. We changed that.

34. Page 21, Lines 9-21: It is not clear whether this paragraph is mostly speculation, or if the authors have performed analysis to convince themselves but are not showing it to the reader. The notion that the latent heat of fusion provides additional buoyancy allowing clouds to penetrate higher has been shown to be incorrect, since the atmosphere adjusts its temperature profile to closely match the temperature profile of convection (Seeley Romps 2016). Moreover, when I look at Figure 5, I see huge CTP anomalies for thick clouds in ALL_LIQ, which are not seen for the other three simulations shown. Is this because the other simulations have few clouds at these large tau values in the mean state, so there is no way of getting a strong altitude feedback, or is the upward shift truly different in the ALL_LIQ case? The change in cloud fraction profile (Figure 7) looks roughly the same in all models, so my sense is that the larger altitude feedback in ALL_LIQ comes from the fact that clouds are optically thicker (higher emissivity) in the mean-state, not something to do with how much the clouds rise in that simulation vs other simulations.

Thanks for your comments. You are right that ALL_LIQ has more thicker clouds in the mean state (as also seen in the vertical profile of cloud liquid water in Figure 2). At

the same time, the change in cloud liquid water profile in the warmer climate is very different in this simulation. This causes the altitude increase in optically thicker clouds as you suggested. We corrected our argumentation.

35. Page 21, line 35: "large increase in cloud top pressure" should be "large increase in cloud altitude feedback" I think.

You are right, we meant "cloud top pressure feedback". We added "feedback".

36. Page 22, first paragraph, also page 24, lines 17-18: I found this paragraph very hard to follow, and it seems like a lot of speculation to me (though not acknowledged as such). First, models run at GCM resolution typically cannot simulate convective aggregation, so it is doubtful that that is playing a role here. Second, Figure 9 (which also lacks a caption) does not really help elucidate the processes described. A zonal mean ΔOLR figure would be a step in the right direction, so the different runs could be compared more easily. I am not aware of anything in Hartmann and Larson (2002) describing convective aggregation or a negative feedback from decreased high cloud coverage. I think the better citation is Bony et al. 2016, which relies on principles from Hartmann and Larson (2002).

On page 22, line 2 of the original manuscript we hypothesize why ECS is not changing, i.e. we acknowledged that we speculate, but we rewrote the paragraph to make that clearer.

In our version, OLR had a caption. Anyhow instead of showing maps of changes in OLR, we now show zonal mean changes of all radiative fluxes in order to provide a complete picture.

You are right that the clustering of convective clouds in the warmer climate is best described by Bony et al. 2016. We added that. We also now refer to an older study with ECHAM which showed a clustering of convection in a simulation in which cirrus scheme had an emissivity of one, i.e. where their infrared optical thickness was highest.

37. Section 6: It is never described how ERFari+aci is computed, and with what type of experiments (fixed SSTs but modified aerosol loading, as in CMIP5?). Can the ari and aci components be separated to get better insights about direct and indirect effects?

As stated above, we removed the entire section on ERFari+aci to make the paper more concise.

38. Page 24, lines 1-9: In and of itself, a large present-day aerosol forcing does not guarantee large TCR or ECS. I think you need to insert words to clarify that "given the observed change in surface temperature and ocean heat uptake, a large present-day aerosol forcing. . ." Similarly, TCR as strictly defined does not depend on ERFari+aci; it is simply the temperature change at the time of doubling for a simulation with CO2 increasing at 1% per year. I think you mean "TCR inferred from observations".

As stated above, we removed the entire section on ERFari+aci to make the paper more concise.

39. Page 24, line 11: Suggest replacing "faster" with something clearer.

As stated above, we removed the entire section on ERFari+aci to make the paper more concise.

40. Page 24, line 12: it is not clear what "that" refers to, or if it is shown in the paper.

As stated above, we removed the entire section on ERFari+aci to make the paper more concise.

41. Page 24, line 32: "variable values" is a little awkward. Also "in contrary" should be "in contrast"

Changed to "calculated values". The spelling has been corrected.

42. Page 24, lines 6-7: Is this Southern Ocean bias really getting to the heart of the

matter, or is it just another symptom of the fundamental issue? My understanding is that SW reflection by clouds in models whose clouds are too optically thin (e.g., due to having too low SLF) is overly sensitive to phase changes because phase changes in these models can actually have a non-negligible effect on cloud optical depth. In contrast, phase changes in in models whose mean-state clouds are not too optically thin have a negligible effect on cloud optical depth. Is this correct, and if not, could this important point be stated more clearly in the paper? It would be nice if this could be shown more unambiguously in the paper. Is it demonstrated anywhere that the models with larger SLFs actually have LWPs that put them in the range of optical depths where albedo is saturated and hence insensitive to phase changes?

Yes, you are right that this problem is not limited to clouds over the Southern Ocean, but there it is most obvious. Albedo only saturates when optical depth exceeds 500 (Kokhanovsky et al., ACP, 2007). Thus, it is not so much the actual saturation than it is the sensitivity of albedo to changes in optical depth that becomes much less at high optical depths.

43. Page 24, lines 8 and 10: "absorbed in clouds" – I don't think this is what you mean

Corrected

44. Page 24, line 32: as far as I can tell, Forster et al (2013) only plots ECS against the total radiative forcing, not ERFari+aci.

As stated above, we removed the entire section on ERFari+aci to make the paper more concise.

45. General comment: It seems that there should be some mention of the various studies finding observational support for a model overestimate of the cloud optical depth feedback at middle latitudes (Gordon Klein 2014, Ceppi et al. 2016, Terai et al. 2016). Currently the Tan et al study is cited alone but it is really one among several studies that are suggesting a bias, one that likely implicates the too-strong phase feedback.

Agreed. We added more references.

---

## Author Comment (AC5) · 4 May 2018

We thank the referee for his/her valuable comments and suggestions. We marked the responses to the comments in red.

General Comments:

It is nice to see that more modeling groups are now looking into the issue of cloud phase and its importance for the overall cloud feedback and climate sensitivity. As such, this paper is a timely and important contribution, but it needs additional work before publication because of the following main reasons:

[Figure]

1) The comparison of supercooled liquid fraction (SLF) with CALIOP, which in many ways forms the foundation for the study, is not done correctly (see comment below).

Sorry, this was written incorrectly. We did do it correctly and now explained it properly.

2) The claim in the abstract that the findings are contrary to those of Tan et al. are not supported by the results. Tan et al. found a systematic change in equilibrium climate sensitivity (ECS) with changing cloud phase. The same relationship between cloud phase and ECS is found here, but ECHAM6 appears to lie elsewhere on the SLF scale (has higher SLF) to begin with than the model used in Tan et al. (CESM). Many of the model experiments presented are not designed to differ from REF predominantly in their cloud phase, so they shed very little light on this relationship. In my opinion, the experiments that are relevant in this respect are ALL_ICE, REF and ALL_LIQ, but for ALL_LIQ and ALL_ICE there is at present not enough information provided on how these simulations were designed.

We added a more thorough description of the design of the ALL_ICE and ALL_LIQ experiments in section 3. You are right that experiments GFAS3.4 and 10/cc were not designed to address ECS, therefore we removed these two experiments. Cloud phase does change profoundly in simulation NOCONV so we kept this simulation. Likewise, we kept simulation HET because a change in ice cloud optical thickness could impact ECS.

In addition to the above comments, I have the following additional ones (chronologically, so minor and major comments are interspersed):

Page 1, line 13: frequent should be frequently

Corrected

Page 1, line 16: Remove one 'is'

Corrected

Page 2, line 2: uncertainty should be uncertainties

Corrected

Page 2, line 15: Emphasize that TCR is most relevant to present-day/modern climate change.

Done

Page 2, lines 9-17: ECS is really the surface temperature change in response to a doubling of $CO_2$ once the fully coupled atmosphere ocean have reached a new equilibrium state. Therefore, simulations with an atmosphere and a MLO are only an approximation of that equilibrium state, and that should be acknowledged here. There are papers that compare the differences in equilibrium climate state depending on whether simulations are run with MLO or full ocean. It is important to point out here that the experiments in this paper differ from those of Tan et al. in this respect.

Point well taken, we now acknowledge both aspects.

Page 2, line 29-30: It wasn't the spread that varied between 1.4 and 4.1 K, but the estimates themselves.

Thanks for noting this. It's now corrected.

Page 2, line 30: Do you mean "somewhat higher" here? Page 2, line 31: positiv should be positive

Both error are corrected.

Page 3, line 3: Maybe clarify that it is the mid-latitude storms that shift poleward?

Added

Page 3, lines 16-24: Tan et al. used the fully coupled CESM model, not just the atmospheric CAM5. Please clarify here what aspect of the climate of the CESM simulations in Tan et al. was not realistic. This was not specified in Sherwood and Gettelman

(2017), so instead of repeating their unsubstantiated claim it would be better to here explain what specifically was unrealistic in those simulations. Tan et al. also did not claim that ECS in CESM was too low (it is actually quite high), but that the cloud phase bias in isolation would bias ECS low. Other biases in CESM could potentially bias ECS high.

Agreed. We now state explicitly that we are referring to the cold bias of 2-3 K in those simulations in Tan et al., that best matched the CALIOP results. This would lead to more ice in the present-day climate and bias the cloud phase feedback high.

Page 5, line 13: What is "cloud blinking"?

Cloud blinking occurred in ECHAM6.1 because of a bug in the cloud cover scheme. The cloud cover was either 0 or 1 in a grid box but almost never any fractional cloud cover was produced. This on and off in cloud cover is called cloud blinking. In later model versions (including ECHAM6.3) this bug was removed and fractional cloud cover in grid boxes is simulated.

Page 7, lines 25-27: How were the ALL_ICE and ALL_LIQ simulations designed? Did you switch off heterogeneous ice nucleation in ALL_LIQ? And in ALL_ICE, did you increase ice nucleation, increase ice crystal growth (through e.g. the WBF process), or both? What about detrained ice/liquid from convection, did you modify that?

We now describe our simulations in more detail.

Page 8, line 3: I'm guessing this should be "horizontal resolution"?

Yes, corrected.

Page 11, line 35 - Page 12, line 1 2: This is a widespread problem in GCMs in general, not just in CAM5/CESM.

You are right. We changed that and added more references.

Page 13, line 35 -Page 14, line 1: This is not the appropriate way to sample the simulated SLF to compare with CALIOP. CALIOP measures SLF for ALL cloud tops, irrespective of their optical depth, but if the cloud top layer has an optical depth < 3 it will ALSO be able to retrieve cloud phase from the cloud layer below cloud top. That is very different from only sampling SLF from optically thin clouds (tau < 3) in the simulations, so the comparison to CALIOP in Fig. 3 is not meaningful at this point.

We actually did it correctly but wrote it wrongly. We corrected that.

Page 14, line 8: Do you mean "detrainment" here? It would actually be very helpful to know exactly how ECHAM handles the cloud phase of detrained clouds.

Yes, we meant detrainment. We now explain how we handle cloud phase from detrainment in section 3.

Fig. 4 and related discussion: The simulations presented here are not comparable to the experiments in Tan et al, for the following reasons: With the exception of ALL_LIQ and ALL_ICE, the differences between REF and the other experiments go far beyond just cloud phase. Whatever difference (or lack of difference) in ECS relative to REF can therefore not be attributed to cloud phase changes alone. In addition, as pointed out above, SLF is not extracted from the simulations in an appropriate manner, so it is hard to say how SLF would compare if they had been, and since it is unclear how ALL_LIQ and ALL_ICE were constructed it is hard to make an informed comparison with the corresponding simulations in Tan et al.

As mentioned above, we did extract SLF correctly, thus the comparison to Tan et al. is justified.

Page 16, line 6: proof should be prove.

Corrected

Table 3: Give two decimals consistently in this table.

We actually prefer to keep the number of decimals according to the value of the quantity. We have 2 decimals for precipitation, where they are needed because the different global mean values are rather similar. For all other quantities 1 decimal is sufficient. To respond to your request, we added one decimal for the oceanic cloud top cloud droplet number concentration.

Fig. 5 shows that there are a lot of other things going on in these simulations other than the cloud phase feedback. Clearly, the ALL_LIQ cloud changes are very different from those in REF, so the fact that they have a similar ECS is the result of multiple compensating differences between them, cloud phase only being one of them.

We agree. We are now discussing this in more detail.

Fig. 7 and related discussion: This figure, if anything, confirms the findings of Tan et al. if you focus on the simulations here that differ ONLY in their cloud phase. The cloud optical depth feedback changes quite dramatically (becomes less negative) from ALL_ICE to REF, and a small additional change is seen from REF to ALL_LIQ. The total change of 0.5K/Wm-2 is very strong. The main difference is that the default ECHAM6 is closer ALL_LIQ in its behavior, while CESM was more similar to the equivalent of ALL_ICE. The other experiments have lots of other changes in them beyond cloud phase and are not relevant for the discussion of the impact of cloud phase on climate sensitivity.

We actually mention that at the very end of section 4. We now discuss that again in the conclusions.

Page 20, line 5: thinnen should be thin

Changed

Page 21, line 3: optical should be optically

Corrected

Page 22, lines 1-2: The only way that a marked increase in the overall cloud feedback

can not correspond to a marked change in ECS is if either: i) The simulations had not equilibrated in the analyzed 25 year time periods, or ii) Other climate feedbacks compensate for the change in the cloud feedback. I'm guessing it's the latter, but both aspects should be addressed in the paper. In other words, what was the radiative balance for the respective experiments for the 25 yr time periods analyzed, and how did the other (non-cloud) climate feedbacks change between the experiments?

Unfortunately, we did not diagnose other feedbacks. What we can say is that our simulations are in equilibrium after 25 years, so that is not the problem. We now discuss the hypothesized compensating feedback in more detail.

Page 24, line 31 -33: These findings are not contrary to Tan et al. - the SLF is higher in ECHAM6 than in CESM/CAM, so there is a smaller ECS increase associated with increasing SLF. It is completely consistent with Tan et al., as far as I can tell. A caveat here is obviously that in the ECHAM6 simulations SLF is not calculated they way it has to be in order to be comparable to CALIOP, so once that's corrected the SLF comparison may look different.

See answers to both points above.

---

## Author Response (AR2)

Dear Holger,

Below please find a point-by-point response to the reviews as well as the marked-up manuscript.

Cheers, Ulrike

**Response to reviewer 1:**

We again would like to thank the referee for his/her valuable comments and suggestions. We marked the responses to the comments in red.

The authors have satisfactorily addressed my comments about the previous draft, with one exception, and the revised manuscript is much improved. I recommend publication after minor revisions to address the points below.

Thanks.

1. The discussion of the Tan et al. (2016) limitations is much improved. They now state that:

Page 3 Line 33 – Page 4 Line 1
"…the results by Tan et al. (2016) should be viewed with caution because their versions of CAM5 that were constrained by the CALIOP observations of SLF, had a 2-3 K colder present-day climate than their reference simulation. This could have impacted ECS because climate feedbacks depend on the state of the system, especially those related to ice, such as the cloud phase feedback and would bias it high."

This is an entirely valid point. However, it is important to show whether or not the model results discussed in this paper have a similar limitation. Without providing information about the temperatures of the runs analyzed in this paper, it is not clear whether or not the same limitation exists with the ECHAM6 results presented here. How do the temperatures of the ALL_LIQ and ALL_ICE simulations compare to REF? Do the model results discussed here move past the limitations of the Tan et al results?

The global mean surface temperatures at 1xCO2 are 15.4ºC in simulation REF, 15.3ºC in simulation ALL_ICE and 16.4ºC in simulation ALL_LIQ. However, given that Tan et al. (2018) showed that the climate sensitivity in CAM does not depend on tuning, we removed this sentence.

2. There is no caption for figure 1 in the revised manuscript.

The figure was too tall so that the caption got dropped. We fixed that.

3. The sentence starting at Line 15 on the abstract "In ECHAMg6-HAM2 the amount…" is quite long and unclear to me. I suggest breaking it up into 2 or more concise statements.

Done

4. Page 15 Line 19. The text seems to imply precipitation rates are listed in Table 3, but I do not see precipitation rates in Table 3.

Good point, we removed the reference to Table 3.

**Response to Trude Storelvmo:**

We thank Trude for her valuable comments and suggestions. We marked the responses to her comments in red.

The authors have addressed many of my previous comments/concerns, and the manuscript is improved over the previous version. A few issues remain, which are listed below. The manuscript should be suitable for publication once these are addressed.

1) Abstract, minor comment: The first two sentences state well-established facts that do not really belong in an abstract. The abstract is also quite long. The abstract could just begin with the 3rd sentence "How clouds change in a warmer climate…"

Changed.

2) Abstract, Line 11: There is nothing in this paper that contradicts Tan et al., so I recommend rewriting this statement. Rather, the study seems to suggest that only severe underestimations (as in the CESM model) of supercooled liquid would significantly suppress climate sensitivity, so I would rather emphasise the novelty of that finding instead of presenting the findings as a contradiction to Tan et al. when they are really not

We think the differences are caused by the hypothesis that you question in your next point, but we agree that contradicting might be too strong a word and changed it to: "different from".

3) Abstract, Line 11-12: There is no support in the paper for this hypothesis, the way it is stated. The simulations do not show that the supercooled liquid fraction is not important.

We are not claiming that the supercooled liquid fraction is not important. Instead we are saying that it matters only in low- and mid-level supercooled clouds which are not shielded by higher lying clouds.

4) Abstract, Lines 19 -21: Without a calculation of the non-cloud feedbacks the study is incomplete - conclusions regarding the importance of various cloud feedback can only be drawn if other feedbacks that are, to various degrees, compensating for the cloud feedbacks are also analysed.

Agreed, we actually now calculate the non-cloud feedbacks. This calculation confirms that the clear-sky non-cloud feedbacks are largest in simulation ALL_LIQ, followed by simulation NO_CONV, i.e. by these simulations which have the largest positive change in net CRE from the radiative kernel method. We added the clear-sky non-cloud feedbacks to Table 3.

5) Page 3, Line 32-35: Tan, Storelvmo and Zelinka (2018) showed that this is not the case, so I suggest removing these sentences.

Done

6) Page 7, Line 30-31: This requires some additional explanation. Why is this necessary, and could this impact the simulated cloud feedbacks and climate sensitivity in ALL_LIQ? ALL_LIQ is by design unrealistic, so why is an unrealistic ice crystal number a concern here? Could you show zonal mean ice crystal number concentrations, perhaps as supplementary information?

We also did a simulation in which we didn't do that. This meant that we had to do much more tuning to get the model back into radiative equilibrium. We added the motivation of keeping tuning to a minimum to the text.

7) Page 15: My previous comment remains: simulations HET and NOCONV are different from the other simulations not mainly because of their cloud phase, but for other reasons. Arguably, switching off convection (NOCONV) is a pretty fundamental change, and its impact on cloud phase would NOT be the most important one. Likewise, the nucleation mechanism in cirrus mainly changes aspects of the simulations OTHER THAN cloud phase. I'm not saying these aren't interesting simulations, but they don't belong in a paper that focuses on the link between cloud phase and climate sensitivity. You need to compare simulations that are designed to be different in their cloud phase ONLY.

We disagree that the focus of this paper is only on the link between cloud phase and climate sensitivity. To better emphasize this point, we changed the title to: "The importance of mixed-phase and ice clouds for climate sensitivity in the global aerosol-climate model ECHAM6-HAM2"

8) Page 23, line 11-13: You need to understand this better. Calculations of the non-cloud feedbacks are much easier to perform than the decomposition of the cloud feedback that you already performed, and they do not require any special output, so it should be doable and would add a lot to the paper. The impression will otherwise be that you didn't follow through in your analysis and therefore do not fully understand what's going on in the simulations.

See response to comment 4. We now add a sentence that refers back to Table 3 that shows that the sum of the clear-sky feedbacks is largest in ALL_LIQ followed by NO_CONV.

[revised manuscript text omitted]